# The use of virtual reality in studying prejudice and its reduction: A systematic review

**Matilde Tassinari**[1]☉*, **Matthias Burkard Aulbach**[1,2]☉, **Inga Jasinskaja-Lahti**[1]

**1** University of Helsinki, Helsinki, Finland, **2** Aalto University, Uusimaa, Finland

☉ These authors contributed equally to this work.

* matilde.tassinari1@gmail.com

**Data Availability Statement:** All relevant data are within the paper and its Supporting Information files.

**Funding:** All authors of this manuscript are funded by Academy of Finland (Suomen Akatemian,

## Abstract

This systematic review provides an up-to-date analysis of existing literature about Virtual Reality (VR) and prejudice. How has VR been used in studying intergroup attitudes, bias and prejudice, are VR interventions effective at reducing prejudice, and what methodological advantages and limitations does VR provide compared to traditional methods are the questions we aim to answer. The included studies had to use VR to create an interaction with one or more avatars belonging to an outgroup, and/or embodiment in an outgroup member; furthermore, they had to be quantitative and peer-reviewed. The review of the 64 included studies shows the potential of VR contact to improve intergroup relations. Nevertheless, the results suggest that under certain circumstances VR contact can increase prejudice as well. We discuss these results in relation to the intergroup perspective (i.e., minority or majority) and target minority groups used in the studies. An analysis of potential mediators and moderators is also carried out. We then identify and address the most pressing theoretical and methodological issues concerning VR as a method to reduce prejudice.

## 1. Theoretical background

Virtual reality (VR) has been used for a variety of purposes, from video games to army training [1] or medical education [2]. In the last decade, it has received increasing attention particularly in social psychology with a range of other applications (see [3] for a meta-analysis), such as prejudice reduction. The great advantage of VR over regular computer-based simulation is the high degree of immersion: it creates a strong illusion of being in the computer-generated world [4]. As VR technology puts participants into a virtual world that can be controlled by the researcher, it allows both further development of known research paradigms with highly standardised experimental conditions and the creation of entirely new paradigms that were not possible before. In this review, we systematically evaluate studies which have utilised VR to study and combat prejudice.

### 1.1 Prejudice, intergroup bias, and intergroup attitudes

Prejudice generally refers to "any attitude, emotion, or behaviour toward members of a group, which directly or indirectly implies some negativity or antipathy toward that group" [5].

www.aka.fi) grant number: 332311 The funders had no role in study design, data collection and analysis, decision to publish, or preparation of the manuscript. Open access funded by Helsinki University Library.

**Competing interests:** The authors have declared that no competing interests exist.

Research in social psychology has studied prejudice within broader frameworks of intergroup bias. In this framework, prejudice is seen as having different manifestations varying from ingroup preference (e.g., ascribing more positive characteristics and emotions to and favouring the ingroup) to out-group derogation (ascribing more negative characteristics, showing more negative or ambivalent emotions to, and discriminating against the outgroup), with both having negative consequences for the outgroup [e.g. 6,7]. Following this approach, we refer to intergroup bias as an overall discrepancy in favour of the ingroup over the outgroup, while discrimination refers to biased behavioral intentions and/or overtly performed behavior that upholds the outgroup's disadvantaged position [e.g. 7].

Prejudice as an intergroup attitude refers, in turn, to evaluative tendencies that can also be studied via three interrelated components [e.g. 8]: a cognitive component (negative stereotyping and attributing negative characteristics to members of the outgroup), an emotional component (e.g., experiencing ambivalent, unpleasant or negative emotions towards outgroups), and a behavioral component (e.g., exhibiting negative behavioral intentions or behaviors such as social exclusion or discrimination against the outgroup).

Another critical distinction divides implicit from explicit intergroup attitudes. While the expression of the latter is normatively monitored, the former are more automatically activated and as such largely defy subjective control [9]–thus having the potential to predict and change behaviours more accurately than their explicit counterpart [see 10 for an overview]. Research on implicit social cognition (i.e. the cognitive processes taking place on the automatic route and involving social groups) has provided researchers with the Implicit Association Test (IAT) [11], which requires subjects to categorise as fast as possible a set of stimuli (e.g. words or faces) into one of two classes of attributes defining evaluations or stereotypes (e.g. good or bad, competent or incompetent). Faster responses indicate a stronger association of the sorted stimuli with the chosen evaluation. Despite severe criticism of the IAT and the concept of implicit attitudes on grounds of unclear validity and methodological problems (see e.g. reviews [12–14]), the IAT and related measures remain popular in studying intergroup bias, especially in socially sensitive topics (see a review by [15]).

Recently, neuroimaging and psychophysiological techniques have also emerged as an alternative way to study intergroup bias by observing their neurological and physiological aetiology and counterparts (see a review by [16]).

## 1.2. Malleability of intergroup attitudes

Prejudice reduction interventions can be divided in several ways based on their theoretical and methodological approaches. Paluck and Green [17] divide social scientific and psychological interventions into those that attempt to influence intergroup cognitions, emotions and behaviours (e.g., social categorisation and identities, intergroup contact and emotions) and those that attempt to support positive intergroup relations via educative (knowledge, critical thinking) or normative influence upon individual emotions or thoughts.

The effectiveness of prejudice-reduction interventions seems to highly depend on the type of social stigmas, outcome measures and the target groups studied. To study the malleability of implicit racial bias, Lai et al. [18] studied a range of 17 different interventions in an experimental design. Their results showed that half (8/17) of the interventions were effective at reducing non-Blacks' implicit preferences for Whites compared to Blacks. In terms of the mechanisms, the most efficient interventions used counter stereotypes and evaluative conditioning methods or provided cognitive and behavioural strategies to override biases, while those inducing perspective-taking (including imagined contact), egalitarian value orientation, or positive emotion turned out to be inefficient. In their study, no intervention consistently reduced explicit

racial preferences, and there were no signs of the extended effect of intervention towards "uncontacted" outgroups. Moreover, the effects of the interventions working with implicit bias also appear to be very short-lived. Lai et al. [19] further examined the stability of the effect of nine interventions and found that although they all immediately reduced implicit preferences, none of them was effective after a delay of several hours to several days.

Beelmann and Heinemann [20] have, in turn, conducted a meta-analysis of 81 studies containing 122 intervention–control comparisons of structured programs to reduce explicit negative intergroup attitudes in children and adolescents via intergroup contact, information/ knowledge acquisition, and promotion of individual social-cognitive competencies. Their results showed that interventions that were based on direct contact experiences and induced empathy and perspective taking showed the strongest effects. The effects also varied according to the program participant's social status, the target out-group, and the outcome measure. Interventions were less effective with emotional and behavioural measures of prejudice than when attempting to change the cognitive components of prejudice; prejudice towards disabled and elderly people was more malleable than towards ethnic minority group members, and attitudes of majority group members towards ethnic minority group members were more prone to the effect of intervention than vice versa (see also [21]). Notably, the majority of studies assessed by Beelmann and Heinemann [20] examined only immediate effects on intergroup attitudes with a mean effect size of around $d = 0.30$. Only ten out of 81 studies evaluated long-term effects and showed stable positive effects or even stronger positive effects over time (e.g., [22]). In this systematic review, we focus on the malleability of explicit and implicit intergroup attitudes via intergroup contact in VR. The volume of intervention studies based on contact theory and the empirical support of the effectiveness of contact-based interventions in prejudice reduction have outlined the potential of this method for prejudice reduction in VR.

## 1.3 Attitude change and malleability via intergroup contact

According to Allport's [23] contact hypothesis, positively engaging with outgroups is a fruitful way to improve intergroup relations. In this paradigm, intergroup contact has been originally defined as actual face-to-face interaction between members of clearly defined groups [24]. Hundreds of studies have demonstrated that intergroup attitudes can be changed and improved by creating positive contact between different groups (see meta-analyses by [3,24,25]). These attitude changes are mainly mediated by increased knowledge about the outgroup, reduced anxiety about intergroup contact, and increased empathy and perspective taking. Importantly, when Allport's [23] conditions for positive intergroup contact (equal status between groups in the situation; common goals; cooperation between groups; support of authorities) are met, prejudice reduction is strongest [24]. On the contrary, interventions can also backfire and lead to increased intergroup conflict if optimal conditions are not met [26].

Moreover, as a considerable corpus of recent research [27] has demonstrated, the positive effect of contact on intergroup attitudes is not limited to direct interactions but also emerges with mediated, extended or indirect contact [24,28,29], including online contact in general [30] and e-contact in particular [31], as well as imagined contact [32].

However, the potential of VR contact to reduce prejudice has not been properly evaluated. Amichai-Hamburger and McKenna [33], as well as Dovidio et al. [27], have stressed that online contact, including VR, might be particularly well suited for creating optimal contact conditions, because it creates an anxiety-safe and controlled environment. According to the meta-analysis by Lemmer and Wagner [3], in which they compared the effects of different direct and indirect forms of contact and which included only eight comparisons with virtual contact, virtual contact intervention programs showed tentative weak evidence for their

usefulness in prejudice reduction. However, the constant development of VR technologies and the accumulating amount of studies utilising VR in prejudice research also continue to improve our understanding of VR as a platform to study and improve intergroup relations, as well as pose demands to evaluate the progress. In addition, there might be several moderators and mediators of the effect of VR contact on intergroup attitudes specific for VR contact. Reaching a better understanding of the features of VR contact can thus help to make it an important avenue for prejudice reduction endeavours.

## 1.4 Virtual reality for prejudice reduction

As previous studies on intergroup contact in VR differ a lot in their approaches and technological solutions, it is important to clarify what we refer to by VR studies and VR intergroup contact. According to Burdea and Coiffet [34], virtual reality is an immersive technology allowing the user to interact in real time with a 3D computer-generated environment simulating reality. One of its defining features is immersion in the environment, which is defined as the sensation of being there [35]. In VR, immersion is allowed by fully experiencing the simulated world through the senses of sight and sound, while the surrounding environment is not visible to the user [36]. This is closely related to the concept of embodiment, which we define by having full control over a virtual avatar, with the avatar's movements being coupled to the movements of one's physical body. This creates the illusion of ownership of the virtual body or perceiving the virtual body representation to be one's own body (see [37]). This sense of embodiment can cause the "Proteus effect", namely a change in people's behaviour and self-representation to match the identity of their virtual self [38,39]. Along with the high degree of immersion, the body ownership illusion allowed by embodiment makes VR users experience a stronger sense of spatial presence compared to the same environment in 2D [40].

How are VR's features serving research on prejudice reduction? Firstly, VR is a unique platform that can be used by researchers both to create and to study intergroup contact, as it enables the experience of direct and indirect intergroup encounters from both majority and minority perspectives. Importantly, VR combines a strong sense of a real social encounter combined with a high degree of experimental control, allowing researchers to ensure optimal conditions of intergroup contact [23]. Specifically, a recent meta-analysis on computer-mediated contact interventions indicated that online contact is typically characterised by a more equal status between groups compared to real-life contact [30], a crucial condition for prejudice-reduction. Given that VR is, on the one hand, computer-mediated but, on the other hand, more realistic than other online encounters, it is important to acknowledge and determine whether VR contact functions the same way or even better than real-world interventions and if so whether the positive effects of VR contact transfer to real-world encounters.

Another aspect that makes VR useful for the study of intergroup processes is that it allows constructing intergroup contact as experienced from both the minority and majority group perspective. VR research achieves this by using embodiment in two ways: either to enable the subject to embody an avatar belonging to the ingroup (usually majority group, as minority respondents have barely been studied), or to embody them in an avatar belonging to the outgroup (most often a stigmatised minority group for the same reason). VR can thus take perspective-taking interventions one step further by allowing the embodiment of avatars representing outgroup members, coming closer to literally taking their perspective.

A widely used alternative to "true" VR is simulating a first-person experience from the point of view of either the ingroup (i.e., majority) or the outgroup (i.e., minority group), without employing an embodied avatar. This kind of virtual experience has a similarly high degree of immersion and realism, but reduced feelings of body ownership. While both perspectives

can be used to simulate intergroup contact in VR, embodiment in a minority avatar can elicit attitude change even without any additional virtual intergroup contact, by allowing the participant to "put themselves in the shoes of an outgroup member", to the extent it is possible to "live" the experience of an outgroup member.

However, given that empathy and perspective-taking have also been shown to have ironic effects in intergroup contexts leading to more helping behaviour and paternalistic attitudes on the expense of willingness to combat prejudice and inequality [41], the question remains, which form of VR contact—intergroup interaction or embodiment of an outgroup member— is more useful in prejudice reduction.

Finally, VR allows studying participants' behaviour in situations, which, for ethical and/or practical reasons, could not be studied in real life settings, such as helping behaviour in emergency situations or intergroup contact with the most vulnerable or isolated populations. Given all of the above reasons for VR's potential, it is thus important to evaluate whether VR is a powerful tool to combat group-specific biases.

## 2. Structure of the systematic review

As the number of studies capitalising on the potential of VR to improve intergroup attitudes increases, the need arises for an overview of research about prejudice and means to combat it in VR. In this systematic review, we draw an exhaustive analysis of existing research describing how virtual reality has been used up to date to study and shape intergroup attitudes through virtual intergroup contact.

Furthermore, we discuss the methodological advantages VR introduces compared to traditional methods and naturalistic interventions, and we seek to provide a critical analysis of the challenges and limitations faced by scholars studying intergroup relations in VR.

As already noted above, most research on prejudice and intergroup contact in general and in VR in particular has so far focused on the majority's attitudes towards members of socially stigmatised groups such as ethnic minorities. In addition, attitudes based on age, disabilities and gender have been widely addressed. However, intergroup relations with different minority groups follow different dynamics as power relations in a society are hierarchically organised so that the prejudice towards some social groups is more normative than towards some others [42]. For example, a meta-analysis of 27 studies with 31 treatment arms by Paluck et al. [28] shows that contact-based interventions directed at ethnic or racial prejudice have generated substantially weaker effects than those targeted towards other social prejudices. For said reasons, in order to assess the potential of VR to reduce prejudice towards different stigmatised groups, we follow the classification of prejudiced or stigmatised groups adopted by Christofi and Michael-Grigoriou [43], which in turn is based on Goffman's [44] categorization of stigma as an individual attribute that causes society to reject those who are affected by it. Thus, we classify studies included in this review based on the type of stigma by which the target outgroup is affected: overt or external deformations (i.e. physical or age-related stigma), deviations in personal traits (i.e. stigmatising behaviours, health status or disorders), and tribal stigmas (i.e. deriving from ethnic or socio-cultural background). Due to the intersectional nature of some stigmatising characteristics (e.g. gender) that lead them to fall into more than one category, we introduce intersectional stigma as a further stand-alone category.

This systematic review is structured as follows: we first describe the method used to review and include eligible studies; then we analyse the results based on the intergroup perspective adopted in the studies, and successively according to the target stigmatised group; we then proceed to review the mediators and moderators that have been investigated to understand the effect of VR contact on prejudice. Next, we provide an overview of the limitations and

advantages of VR for prejudice reduction. We lastly discuss and summarise the findings and address future research.

Subsequently, we lay out the following research questions:

RQ1: How has VR been used to study prejudice towards stigmatised minority groups?

RQ2: How effective are VR interventions based on intergroup contact at reducing prejudice?

RQ3: Which features of VR, contextual, or individual factors mediate the effect of VR contact on prejudice? (mediators)

RQ4: Under which conditions does VR contact influence prejudice? (moderators)

RQ5: What methodological advantages and limitations characterise VR contact compared to traditional forms of intergroup contact?

## 3. Method

Before beginning our literature search, we pre-registered this review in the PROSPERO database (https://www.crd.york.ac.uk/prospero/display_record.php?RecordID=222294). We report all deviations from the pre-registered plan in this review article.

### 3.1. Information sources and search strategy

Based on the field of interest (i.e., intergroup contact and prejudice in VR), we developed 16 search terms related to virtual reality and intergroup relations (the final search strategy can be found at https://mfr.de-1.osf.io/render?url=https://osf.io/rp3wg/?direct%26mode=render%26action=download%26mode=render) and used these to search published studies in three electronic databases (PsycInfo, Scopus, and Web of Science). The search terms were the following: (Vr OR virtual reality OR immersive virtual environment OR simulation-based assessment OR virtual reality exposure therapy OR virtual OR augmented reality) AND (intergroup relations OR ingroup outgroup OR prejudice OR discriminat* OR bias OR stereotyp* OR stigma* OR intergroup attitude* OR outgroup attitude*).

Searches were limited to human samples and articles published in English, German, Finnish, or Italian. The original search was conducted in March 2021 and updated and expanded during the revision process in January 2022. Once duplicates were excluded, 15,504 citations remained for screening. After title and full-text screening, we searched the reference sections of all included articles for further eligible studies, and contacted all authors of included studies for further articles and unpublished data. The search resulted in 64 studies (51 published journal articles, 11 conference papers, and two dissertations), of which 4 were provided by authors. For more detailed information about the search and screening process, see the PRISMA flow diagram reported in Fig 1.

### 3.2. Inclusion and exclusion criteria

After initial search, we carefully investigated the methods used in each study using the following criteria. Included studies had to use immersive virtual reality (IVR; see e.g. [4]), that is participants had to wear a head-mounted display, be in a room that was "transformed" into a virtual environment with projectors, or use a device to induce augmented reality (we had not pre-registered augmented reality studies but decided to include them due to the high degree of similarity). We excluded qualitative studies and opinion pieces. Moreover, we did not include studies that presented the virtual content on a computer screen. In the virtual reality, participants had to either embody an avatar representing the outgroup or take the perspective of a



## PRISMA 2009 Flow Diagram

**Identification**

Records identified through
database searching
(n =17.011)

Additional records identified
through other sources
(n = 4)

**Screening**

Records after duplicates removed
(n = 13.147)

Abstracts screened
(n = 265)

Records excluded
(n = 190)

**Eligibility**

Full-text articles assessed
for eligibility
(n = 167)

Full-text articles excluded
(n = 103)
Reasons for exclusion:
- VR was not used (n=65)
- Absence of outcomes related to prejudice (n=17)
- No empirical data provided (n=12)
- Other reasons (n=9)

**Included**

Studies included in
qualitative synthesis
(n = 64)

*From:* Moher D, Liberati A, Tetzlaff J, Altman DG, The PRISMA Group (2009). *P*referred *R*eporting *I*tems for *S*ystematic Reviews and *M*eta-*A*nalyses: The PRISMA Statement. PLoS Med 6(7): e1000097. doi:10.1371/journal.pmed1000097

**For more information, visit www.prisma-statement.org.**

**Fig 1. PRISMA flow diagram.**

social group they did not belong to, and/or get in contact with at least one member of an outgroup.

In case of intervention studies, control groups would either receive an intervention to reduce prejudice other than in virtual reality (e.g. real-life interaction, perspective taking exercise, using non-immersive technology), or they would embody or interact with virtual ingroup members (i.e. only intragroup, but no intergroup contact), or would not experience any kind of intergroup contact.

Studies had to report at least one measure of intergroup bias, such as different measures of implicit and explicit prejudice, stereotypes, physiological measures associated with prejudice, or behavioural measures such as physical proximity.

### 3.3. Data extraction

Two independent researchers (MT and MA) conducted all literature searches, screened titles, abstracts, and full-text articles. Selection was such that after each step, any title or abstract that was deemed relevant by either researcher was included in the next step. Both authors then agreed on the final set of included articles. In case of disagreement ($k$ = 8 articles), a third author (IJ-L) made the final decision. The final set of included articles constituted 64 studies reported in 62 independent articles.

Extracted data included publication year and language, study design (within or between participants), country of research site, sample characteristics (size, average age, gender composition, ethnicity), the VR medium and apparatus (VR headset, augmented reality, virtual world projected into a room), the group that was the prejudice target, how outgroups were represented (3D video, virtual agents, avatars, embodying an outgroup avatar), how the contact was designed, the intergroup bias measure, examined hypothesised mediators (such as empathy, gratitude, inclusion of other in the self) and moderators (such as socio-economic status). The extracted data can be seen in Table 1.

We applied the Cochrane Collaboration's tool [45] to assess risk of bias for all included studies. The risk of bias in each study was judged as high, low, or unclear, on each of the following domains: selection bias, performance bias (on experimenter and participant level), detection bias, attrition bias (on participant level as well as outcome level), and reporting bias. Both authors independently assessed the risk of bias on all studies and disagreements were solved through discussion between the coders. Appendix 1 provides an overview of the distribution of risks of bias (low, unclear, high) across studies.

## 4. Results

### 4.1. Descriptive results: Studying prejudice in VR

We found altogether 64 eligible original studies. A list of the included studies can be found in Table 1. Of the 64 studies included in the review, 10 are observational, that is, their main aim is to assess intergroup attitudes in VR, with the remaining studies delivering interventions aimed at decreasing participants' prejudice. All of the 10 observational designs fall into the majority perspective classification, and either focus on tribal stigma (7 studies) or intersectional stigma (3 studies). Only four of the included studies encompass longitudinal measures. The majority ($k$ = 38) of included studies use a between-subjects design, 16 studies a within-subjects design, eight studies opt for a mixed within-between subjects designs, and finally two studies did not employ any control condition.

It is worth pointing out that 32 of the included studies, accounting for half of the total number, was published between 2020 and 2022, showing the growing interest and the rapid

**Table 1. List of included studies.**

| Title | Authors | Year | Country of data collection | Total N | Percent female | Ethnic composition | Sample age | Type of design | Control condition | Prejudice target | Kind of interaction | Prejudice-related measures | Examined mediators | Examined moderators | Main results |
|---|---|---|---|---|---|---|---|---|---|---|---|---|---|---|---|
| Contact in VR: Testing Avatar Customisation and Common Ingroup Identity Cues on Outgroup Bias Reduction | Alvidrez, S. & Peña, J. | 2020 | USA | 135 | 82.3 | 52.5% Asian; 20.6% Hispanic; 17% Caucasian; 9.9% other | 20.47 (sd = 2.05) | between | avatar self-resembling vs not and common ingroup identity cues vs not | Hispanics in the US | avatar | social distance scale | engagement presence scale | common ingroup identity (university) | Having a self-resembling avatar resulted in decreased engagement presence and higher engagement presence was linked to larger social distance. Common ingroup identity cues had no effect on outcomes. |
| Verbal Mimicry Predicts Social Distance and Social Attraction to an Outgroup Member in Virtual Reality | Alvidrez, S. & Peña, J. | 2020 | USA | 54 | 87 | 54.1% Asian, 18.5% Hispanic, 18.5% Caucasian,9% other | N/A (18–32) | between | avatar self-resembling vs not and common ingroup identity cues vs not | Hispanics in the US | avatar | social distance scale | verbal mimicry | common ingroup identity (university) | Neither avatar customization nor a common ingroup identity predicted verbal mimicry in VR interactions with a Hispanic outgroup member. Verbal mimicry predicted social attraction positively and social distance negatively. |
| Virtual body ownership and its consequences for implicit racial bias are dependent on social context | Banakou, D., Beacco, A., Neyret, S., Blasco-Oliver, M., Seinfeld, S., & Slater, M. | 2020 | Spain | 92 | 100 | 100% white | 21.8 | between-subjects | Embodying a white virtual body | black people | embodiment | black-white IAT; 'attitudes to Blacks' questionnaire | N/A | positivity/negativity of interaction | Negative experiences while embodying a Black avatar can lead to worse implicit attitudes towards Black people in White participants. |
| Virtual Embodiment of White People in a Black Virtual Body Leads to a Sustained Reduction in Their Implicit Racial Bias | Banakou, D., Hanumanthu, P. D., & Slater, M. | 2016 | Spain | 90 (in 2 studies) | 100 | 100% white | 21.9 | between-subjects | Embodying a white virtual body | black people | embodiment | black-white IAT | N/A | number of exposures | Practicing Tai Chi while embodying a Black avatar can reduce White participants' implicit bias against Black people. |
| Virtually Being Einstein Results in an Improvement in Cognitive Task Performance and a Decrease in Age Bias | Banakou, D., Kishore, S., & Slater, M. | 2018 | Spain | 30 | 0 | N/A | 22 | between-subjects | Embodying a young male adult body, i.e. similar to the subjects' own bodies | elderly people | embodiment | age IAT | N/A | N/A | Participants who embodied an avatar that looked like older Einstein performed better on a cognitive task and showed reduced implicit bias against older people, compared to a control condition. |

(*Continued*)

**Table 1.** (Continued)

| Title | Authors | Year | Country of data collection | Total N | Percent female | Ethnic composition | Sample age | Type of design | Control condition | Prejudice target | Kind of interaction | Prejudice-related measures | Examined mediators | Examined moderators | Main results |
|---|---|---|---|---|---|---|---|---|---|---|---|---|---|---|---|
| Racial bias and in-group bias in virtual reality courtrooms. | Bielen, S., Marneffe, W., & Mocan, N. | 2021 | Belgium | 275 | N/A | N/A | N/A | within | white vs non-white defendants | non-white Belgians | 3D video | convictions and sentence harshness in a trial | Perception of terrorism being a very important problem | N/A | Conviction rates are higher for defendant minority member defendant than for white defendants in 3D video staged trials, regardless of the evaluator's group membership. In terms of sentence harshness there is ingroup bias overall. |
| Presence, what is is good for? Exploring the benefits of virtual reality at evoking empathy towards the marginalized | Boehm, N. | 2020 | USA | 199 | 66 | 84% White | N/A | between | VR vs desktop version vs mere perspective-taking vs imagination exercises | drug users | 3D video | drug user stereotypes | Empathy: Interpersonal Reactivity Index scale | presence was positively related to empathy | A VR perspective taking intervention led to stronger feelings of physical presence than a desktop version. Feelings of presence were correlated with empathy towards drug users but did not differ between conditions. |
| Reducing Outgroup Bias through Intergroup Contact with Non-Playable Video Game Characters in VR | Breves, P. | 2020 | Germany | 86 | 50 | 100% White | 20.9 | between-subjects | Helping a Black NPC in a non-VR video game OR Helping a white confederate | black people | virtual agent | black-white IAT; explicit prejudice against Blacks (Pettigrew & Meertens, 1995) | N/A | N/A | Helping a Black video game character reduced explicit but not implicit bias towards Black people and more so when the game was played in a VR compared to desktop version. |
| Perspective-Taking in Virtual Reality and Reduction of Biases against Minorities (Study 1) | Chen, V., Chan, S. & Tan, Y. | 2021 | Singapore | 71 | 46 | 100% Singaporean Chinese | 24.28 (sd = 1.75) | between/mixed | comparison between affective and cognitive instructions | Malay Singaporean | embodiment | feeling thermometers towards ingroup and outgroup | self-other overlap; empathic feelings | N/A | Experiencing an ethnic discrimination scene from the outgroup perspective led to less ingroup bias by decreasing ingroup attitudes, independently from whether participants received a cognitive or affective perspective-taking instruction. |
| The Effect of VR Avatar Embodiment on Improving Attitudes and Closeness Toward Immigrants | Chen, V., Ibasco, G., Leow, V., & Lew, J. | 2021 | Singapore | 171 | 58.5 | 100% Singaporean Chinese | 22.43 (sd = 2.07) | between | ingroup embodiment | PRC Chinese in Singapore | embodiment | feeling thermometer towards target outgroup | empathy; Self-Other overlap | social identity orientation | Experiencing ethnicity-based discrimination embodying an immigrant outgroup avatar improves attitudes and closeness towards that group. |

*(Continued)*

**Table 1.** (Continued)

| Title | Authors | Year | Country of data collection | Total N | Percent female | Ethnic composition | Sample age | Type of design | Control condition | Prejudice target | Kind of interaction | Prejudice-related measures | Examined mediators | Examined moderators | Main results |
|---|---|---|---|---|---|---|---|---|---|---|---|---|---|---|---|
| A Virtual Reality Simulation of Drug Users' Everyday Life: The Effect of Supported Sensorimotor Contingencies on Empathy | Christofi, M., Michael-Grigoriou, D., & Kyrlitsias, C. | 2020 | Cyprus | 40 | 52.5 | N/A | N/A | between-subjects | same content presented on a desktop computer | Drug users | embodiment | Attitudes towards drug users | Empathy; Inclusion of Other in the Self (IOS) scale | Interpersonal reactivity index (IRI) | Self-reported attitudes towards drug users improved after experiencing different situations from a drug user's perspective, both in a VR and a desktop application. |
| VR Disability Simulation Reduces Implicit Bias Towards Persons With Disabilities | Chowdhury, T., Ferdous, S., & Quarles, J. | 2021 | USA | 71 | 39 | N/A | 20.3 (sd = 4.6) | between | 2X2 design (interface: wheelchair vs gamepad; immersion: VR vs desktop) | People with disabilities | embodiment | IAT towards with people with disabilities | N/A | N/A | Embodying a person in a wheelchair in VR with a wheelchair interface reduced implicit bias against people with disabilities more than a desktop version and a gamepad. |
| A Wheelchair Locomotion Interface in a VR Disability Simulation Reduces Implicit Bias | Chowdhury, T. & Quarles, J. | 2021 | USA | 40 | 35 | N/A | 23.6 (sd = 3.1) | between | no wheelchair interface | People with disabilities | embodiment | IAT towards with people with disabilities | N/A | Narrator abled vs disabled | Self-reported attitudes towards drug users improved after experiencing different situations from a drug user's perspective, both in a VR and a desktop application. |
| Influence of weight etiology information and trainee characteristics on Physician-trainees' clinical and interpersonal communication. | Cohen, R. W., & Persky, S. | 2019 | USA | 119 | 52 | 55.5% White, 23.5% Asian, 21% Black, 3.4% Hispanic | 26.3 | between-subjects | Reading an article unrelated to weight | People with obesity | virtual agent | use of stigmatizing language; responsiveness to patient information needs; communication length; lifestyle counseling; lifestyle assumptions | N/A | N/A | Physician trainees were more likely to talk about weight with and provide lifestyle counselling to virtual patients with obesity when they had just read articles about behavioural or genetic influences on weight relative to a control condition. In the behavioural condition, they also used more stigmatizing language and made more assumptions about patients' lifestyle. |
| Using virtual reality to induce gratitude through virtual social interaction | Collange, J., & Guegan, J. | 2020 | France | 80 | 61 | N/A | 21 | between-subjects | Interaction with an ingroup avatar | black people (Study 1) | avatar | social support intentions | gratitude; impression formation (warmth and competence) | N/A | Receiving help from a virtual outgroup avatar increased participants' willingness to offer social support to benefactors and this effect was mediated by perceived warmth. |

(*Continued*)

**Table 1.** (Continued)

| Title | Authors | Year | Country of data collection | Total N | Percent female | Ethnic composition | Sample age | Type of design | Control condition | Prejudice target | Kind of interaction | Prejudice-related measures | Examined mediators | Examined moderators | Main results |
|---|---|---|---|---|---|---|---|---|---|---|---|---|---|---|---|
| Prosocial Virtual Reality, Empathy, and EEG Measures: A Pilot Study Aimed at Monitoring Emotional Processes in Intergroup Helping Behaviors | D'Errico, F., Leone, G., Schmid, M., & D'Anna, C. | 2020 | Italy | 40 | 47.5 | 100% White | 23.8 | between-subjects | Interaction with an ingroup member | black people | 3D video | empathy | EEG-measured calmness, engagement, alertness | social appearance | White participants showed stronger self-reported and neurophysiological stress reactions as well as empathy when interacting in a helping situation with a White person with beggars' clothes or a Black person in business attire than in the other conditions. |
| Reducing the schizophrenia stigma: A new approach based on augmented reality. | de C. Silva, R. D., Albuquerque, S. G. C., de V. Muniz, A., Reboucas Filho, P. P., Ribeiro, S., Pinheiro, P. R., & Albuquerque, V. H. C. | 2017 | Brazil | N/A | N/A | N/A | N/A | no control group | N/A | schizophrenia patients | augmented reality | Questionnaire about stigma | N/A | N/A | Medical students showed increased empathy, pity, fear, stigma, and willingness to help towards patients with schizophrenia after using an augmented reality tool to simulate schizophrenia symptoms. |
| Virtual prejudice | Dotsch, R., & Wigboldus, D. H. J. | 2008 | Netherlands | 33 | 63.6 | 100% white | 21.4 | within-subjects | Interaction with an ingroup avatar | Moroccans (in the Netherlands) | virtual agent (but not explicitly stated; remains passive) | single-target IAT with Moroccan names; explicit prejudice; distance from outgroup avatar | skin conductance | N/A | Participants kept more distance from a virtual agent with Moroccan vs White features, which was predicted by implicit attitudes towards Moroccans. The effect was mediated by skin conductance levels. |
| The behavioral dynamics of shooter bias in virtual reality: The role of race, armed status, and distance on threat perception and shooting dynamics | Eiler, B. A. | 2017 | USA | 61 | N/A | N/A | N/A | within-subjects | Interaction with an ingroup avatar | Black people | virtual agent (but not explicitly stated; remains passive) | race-weapon IAT; shooter bias | heart rate; perceived threat | distance from outgroup member and armed status (gun, phone, no object) | Black virtual agents were perceived as more threatening and shot at more often than White virtual agents in a shooter bias paradigm. |
| Virtual Virgins and Vamps: The Effects of Exposure to Female Characters' Sexualized Appearance and Gaze in an Immersive Virtual Environment | Fox, J., & Bailenson, J. | 2009 | USA | 83 | 48 | 38.6% White; 24.1% Asian/Asian-American; 13.3% Black/African-American; 10.8% Latino/Latina/Hispanic; 13.3% multiracial | 20.82 (sd = 3.17; range = 18–34) | between | N/A | Women | virtual agent | Ambivalent Sexism Inventory | N/A | N/A | Participants self-reported stronger sexism after perceiving stereotypical vs counter-stereotypical female virtual avatars. |

*(Continued)*

**Table 1.** (Continued)

| Title | Authors | Year | Country of data collection | Total N | Percent female | Ethnic composition | Sample age | Type of design | Control condition | Prejudice target | Kind of interaction | Prejudice-related measures | Examined mediators | Examined moderators | Main results |
|---|---|---|---|---|---|---|---|---|---|---|---|---|---|---|---|
| The Effect of Embodying a Woman Scientist in Virtual Reality on Men's Gender Biases | Freedman, G., Green, M.C., Seidman, M., Flanagan, M. | 2021 | USA | 96 | 0 | .3% African American or Black, 1.0% Arab/Middle Eastern, 31.3% Asian, Asian American, or Asian Canadian, 3.1% Hispanic/Latino, 47.9% White, 7.3% Multiracial, 2.1% other | 19.79 (sd = 1.63) | between/ mixed | embodying a male virtual body | women | embodiment | Gender-Science IAT; explicit attitudes towards women in STEM (perceptions of the climate for women in STEM, stereotype endorsement) | N/A | N/A | Participants didn't show any improvement in implicit or explicit bias when embodying a female scientist avatar, compared to male avatars performing the same tasks. Game enjoyment was also unvaried between condition. There was no interaction between condition and reveal (early vs. late avatar gender reveal) on the main outcomes. Exploratory analyses showed that participants felt more positive emotions after playing. |
| Psychological response to an emergency in virtual reality: Effects of victim ethnicity and emergency type on helping behavior and navigation | Gamberini, L., Chittaro, L., Spagnolli, A., & Carlesso, C. | 2015 | Italy | 96 | 50 | 100% White | 24 | between-subjects | Interaction with an ingroup avatar | Black people | virtual agent | helping behaviour in an emergency situation | N/A | N/A | In a virtual helping situation, White participants were less likely to help Black than White virtual agents under time pressure, but no differences were found in dangerous situation (fire in a building). |
| Being the Victim of Intimate Partner Violence in Virtual Reality: First-Versus Third-Person Perspective | Gonzalez-Liencres, C., Zapata, L. E., Iruretagoyena, G., Seinfeld, S., Perez-Mendez, L., Arroyo-Palacios, J., Borland, D., Slater, M., & Sanchez-Vives, M. V | 2020 | Spain | 32 | 0 | N/A | 32 | between-subjects | N/A | women | embodiment | gender IAT | N/A | N/A | Men who experienced a situation of intimate partner violence while embodied in a female avatar showed stronger physiological and behavioural reactions and reported stronger feelings of identification with the victim, and of taking the scene personally than a third-person perspective control group. |
| The influence of racial embodiment on racial bias in immersive virtual environments | Groom, V., Bailenson, J. N., & Nass, C. | 2009 | USA | 98 | 60 | 45.1 White, 21 Asian, 15.6 Black, 6.9 Hispanic, 7.8 Other | N/A | between-subjects | Perspective-taking exercise (imagining a day in the life of Black person) | Black people | embodiment | black-white IAT; interpersonal distance; Racial Argument Scale; Modern Racism Scale | N/A | N/A | Participants showed stronger implicit bias against Black people after embodying a Black compared to White avatar. |

*(Continued)*

**Table 1.** (Continued)

| Title | Authors | Year | Country of data collection | Total N | Percent female | Ethnic composition | Sample age | Type of design | Control condition | Prejudice target | Kind of interaction | Prejudice-related measures | Examined mediators | Examined moderators | Main results |
|---|---|---|---|---|---|---|---|---|---|---|---|---|---|---|---|
| Virtual Humans and Persuasion: The Effects of Agency and Behavioral Realism | Guadagno, R. E., Blascovich, J., Bailenson, J. N., & McCall, C. | 2007 | USA | 65 (study 1) + 174 (study 2) | 45 (study 1); 51 (study 2) | N/A | N/A | between-subjects | Interaction with male virtual agents | Women | virtual agent | susceptibility to persuasion; dimensions of person perception | N/A | N/A | Participants changed their attitudes more towards a virtual agent's attitude when the virtual agent of the same gender as the participant rather than with a virtual agent of the other gender. |
| Conceptual knowledge and sensitization on Asperger's syndrome based on the constructivist approach through virtual reality | Hadjipanayi, C., & Michael-Grigoriou, D. | 2020 | Cyprus | 40 | 50 | N/A | N/A | between-subjects | Reading a text about Asperger's syndrome | People with Asperger's syndrome | embodiment | empathy; sensitization towards Asperger's syndrome | N/A | N/A | Participants who used a VR-based simulation of Aspergers gained more knowledge about the syndrome than participants who read about Aspergers. |
| Virtual race transformation reverses racial ingroup bias | Hasler, B. S., Spanlang, B., & Slater, M. | 2016 | Spain | 32 | | | | within-subjects | embodying an ingroup avatar | Black people | embodiment | black-white IAT; liking of the other person; mimicry | N/A | N/A | White participants mimicked virtual agents more if their virtual avatar was of the same skin colour as the virtual agent's than if there was discordance in skin colour, regardless of participants' implicit race bias. |
| Virtual Peacemakers: Mimicry Increases Empathy in Simulated Contact with Virtual Outgroup Members | Hasler, B. S., Hirschberger, G., Shani-Sherman, T., & Friedman, D. A. | 2014 | Israel | 57 | 100 | 100% White | N/A | mixed | counter-mimicking an outgroup virtual agent | Palestinians | virtual agent | empathy; sympathy; self-other overlap; interaction harmony; outgroup affect | N/A | N/A | Conversing with a virtual agent representing a Palestinian led to increased empathy, sympathy, more felt closeness, and to perceptions of a more harmonious interaction when the virtual agent mimicked the participant compared to counter-mimicry. |

*(Continued)*

**Table 1.** (Continued)

| Title | Authors | Year | Country of data collection | Total N | Percent female | Ethnic composition | Sample age | Type of design | Control condition | Prejudice target | Kind of interaction | Prejudice-related measures | Examined mediators | Examined moderators | Main results |
|---|---|---|---|---|---|---|---|---|---|---|---|---|---|---|---|
| Virtual Reality-based Conflict Resolution: The Impact of Immersive 360° Video on | Hasler, B., Hasson, Y., Landau, D., Eyal, N. S., Giron, J., Levy, J., Halperin, E., & Friedman, D | 2020 | Israel | 100 | 0 | 100% Jewish Israeli | 25.4 | between-subjects | Watching a video on a desktop computer | Palestinians | 3D video | moral justification of soldiers' actions | perspective-taking; empathic emotions; hostile emotions; skin conductance | N/A | Jewish-Israeli participants who watched a 360° video of a conflict scenario between Israeli soldiers and a Palestinian couple judged the soldiers' actions to be less moral and less justified and reported more hostile emotions towards the soldiers relative to a group who watched the video in 2D. These effects were mediated by a higher sense of presence and engagement in the 360° video. |
| The enemy's gaze: Immersive virtual environments enhance peace promoting attitudes and emotions in violent intergroup conflicts (Study 1) | Hasson, Y., Schori-Eyal, N., Landau, D., Hasler, B. S., Levy, J., Friedman, D., & Halperin, E. | 2019 | Israel | 112 | 71 | 100% Jewish Israeli | 24.3 | between-subjects | Taking the ingroup's perspective in the 360° video | Palestinians | 3D video | empathy; fear of the targets; positive appraisals; Attribution of future benign intentions support for economic compensation | N/A | N/A | Jewish-Israeli participants who watched a 360° video of a conflict scenario between Israeli soldiers and a Palestinian couple from the outgroup's point of view perceived Palestinians more positively than those who watched the scene from the ingroup point of view. |
| The enemy's gaze: Immersive virtual environments enhance peace promoting attitudes and emotions in violent intergroup conflicts (Study 2) | Hasson, Y., Schori-Eyal, N., Landau, D., Hasler, B. S., Levy, J., Friedman, D., & Halperin, E. | 2019 | Israel | 100 (55 at follow-up 5 months later) | 77 | 100% Jewish Israeli | 23.9 | between-subjects | Taking the ingroup's perspective in the 360° video | Palestinians | 3D video | empathy; fear of the targets; dehumanization; perceived threat; shoot/no-shoot dilemmas | N/A | N/A | Jewish-Israeli participants who watched a 360° video of a conflict scenario between Israeli soldiers and a Palestinian couple from the outgroup's point of view perceived Palestinians more positively and judged a real-world ingroup transgression five months later more harshly than those who watched the scene from the ingroup point of view. |

(*Continued*)

**Table 1.** (Continued)

| Title | Authors | Year | Country of data collection | Total N | Percent female | Ethnic composition | Sample age | Type of design | Control condition | Prejudice target | Kind of interaction | Prejudice-related measures | Examined mediators | Examined moderators | Main results |
|---|---|---|---|---|---|---|---|---|---|---|---|---|---|---|---|
| The effect of gender, religiosity and personality on the interpersonal distance preference: a virtual reality study | Hatami, J., Sharifian, M., Noorollahi, Z., & Fathipour, A. | 2020 | Iran | 46 | 71 | 100% Iranian | 23.8 | between-subjects | N/A | Gender | 3D video | preferred distance from the outgroup target member | N/A | N/A | Viewing actors in a 360° video, religious Iranian participants preferred further distances between themselves and opposite gender actors compared to non-religious Iranian participants. |
| The Virtual Doppelganger—Effects of a Virtual Reality Simulator on Perceptions of Schizophrenia | Kalyanaraman, S. S., Penn, D. L., Ivory, J. D., & Judge, A. | 2010 | USA | 112 | 52 | N/A | 22.2 | between-subjects | empathy-inducing instructions without use of VR | People with Schizophrenia | augmented reality | Empathic feelings for people suffering from schizophrenia; social distance scale; attitudes | N/A | N/A | Participants who underwent an augmented reality simulation of schizophrenia with an instruction to empathize with schizophrenia patients showed increases in empathy with and more positive perceptions of people with schizophrenia. Using the augmented reality apparatus without empathy instructions resulted in a stronger desire to keep social distance from people with schizophrenia. |
| Processing Racial Stereotypes in Virtual Reality: An Exploratory Study Using Functional Near-Infrared Spectroscopy (fNIRS) | Kim, G., Buntain, N., Hirshfield, L., Costa, M. R., & Chock, T. M. | 2019 | USA | 13 | 47 | 53.8 White, 23.1 Hispanic, 23.1 Asian | N/A | Within-subjects | racially charged scene vs holiday scene | Black people | 3D video | Brain activation in mPFC, right lPFC, left lPFC | N/A | N/A | Viewing a racially-charged animated scene in VR led to stronger brain activation in the right and left lateral prefrontal cortex than viewing a holiday scene, indicating stronger stereotype activation and suppression in the racially charged scene. |

(*Continued*)

**Table 1.** (Continued)

| Title | Authors | Year | Country of data collection | Total N | Percent female | Ethnic composition | Sample age | Type of design | Control condition | Prejudice target | Kind of interaction | Prejudice-related measures | Examined mediators | Examined moderators | Main results |
|---|---|---|---|---|---|---|---|---|---|---|---|---|---|---|---|
| A Virtual Reality Embodiment Technique to Enhance Helping Behavior of Police Towards a Victim of Police Racial Aggression | Kishore, S., Spanlang, B., Iruretagoyena, G., Halan, S., Szostak, D., Slater, M. | 2021 | USA | 38 | 17 | N/A | N/A | within | racially charged scene vs holiday scene | Black people | 3D video | behavioral data based on ratings of participant actions and words (helping behaviour towards the victim) | N/A | N/A | After witnessing an abusive questioning of an African American suspect in VR, US police officers were embodied in two different conditions, witness or victim. 3–4 weeks later, they experienced another abusive episode towards an African American person in a cafe, while embodying a White police officer. Participants in the victim condition showed greater support for the victim in the café situation. |
| Testing an Immersive Virtual Environment for Decreasing Intergroup Anxiety among University Students: An Interpersonal Perspective | Kuuluvainen, V., Virtanen, I., Rikkonen, L., Isotalus, P. | 2021 | Finland | 50 | 78 | 98% Finnish; 2% Finnish-Russian | 27.6 (sd = 7.84; range = 19–49) | between | observer condition (no embodiment) | Middle-Eastern people | embodiment | Intergroup anxiety survey | N/A | N/A | watching a 3D video of a Middle-Eastern man talking about his life and interacting with his family decreases intergroup anxiety in participants, but there is no difference compared to the control group, which was exposed to the same video in 2D. |
| No Country for Old Men? Reducing Ageism Through Virtual Reality Embodiment | La Rocca, S., Brighenti, A., Tosi, G., & Daini, R. | 2019 | Italy | 24 | 50 | N/A | 23.7 | Within-subjects | watched 30 years old hand being tapped; anatomical vs non-anatomical arm position | elderly people | 3D video | Fraboni Ageism Scale; Age IAT | N/A | N/A | Young adults showed decreased implicit age bias after having embodied a virtually old body whose arm was touched in synchrony with their own physical arm. |
| How Does Embodying a Transgender Narrative Influence Social Bias? An Explorative Study in an Artistic Context | Lesur, M. R., Lyn, S., & Lenggenhager, B. | 2020 | Switzerland | 114 | 36 | N/A | 34.1 | between-subjects | VR experience without transgender narrative | Transgender people | 3D video | attitudes towards transgender people; IAT (short version) | N/A | N/A | Embodying a Transgender body in 360° video with or without synchronous tactile stimulation did not change implicit transgender bias. |

(*Continued*)

**Table 1.** (Continued)

| Title | Authors | Year | Country of data collection | Total N | Percent female | Ethnic composition | Sample age | Type of design | Control condition | Prejudice target | Kind of interaction | Prejudice-related measures | Examined mediators | Examined moderators | Main results |
|---|---|---|---|---|---|---|---|---|---|---|---|---|---|---|---|
| Humans adjust virtual comfort-distance towards an artificial agent depending on their sexual orientation and implicit prejudice against gay men | Lisi, M.P., Fusaro, M., Tieri, G., Aglioti, S.M. | 2021 | Italy | 72 | 50 | N/A | N/A (18-35) | no control | no control | gender | virtual agent | interpersonal distance, explicit sexual prejudice and IAT | N/A | sexual orientation, gender, | Heterosexual Men chose a larger distance toward the male avatar compared to Non-Heterosexual Men; also, among women, the heterosexual participants chose a larger distance toward the female avatar compared to the non-heterosexual ones. |
| Investigating Implicit Gender Bias and Embodiment of White Males in Virtual Reality with Full Body Visuomotor Synchrony | Lopez, S., Yang, Y., Beltran, K., Kim, S.J., Hernandez, J. C., Simran, C., Yang, B., & Yuksel, B. F. | 2019 | USA | 24 | 0 | 100% White | 29.8 | mixed | Embodiment in an ingroup avatar | Women | embodiment | gender IAT | N/A | N/A | Male participants who practiced Tai-Chi in a female virtual body showed increases in implicit gender bias while those doing the same exercise with a male virtual body did not show significant changes. |
| Mitigating Negative Effects of Immersive Virtual Avatars on Racial Bias | Maloney, D. | 2018 | USA | 26 | 0 | 100% White | N/A | mixed | Embodiment in an ingroup avatar | Black people | embodiment | IAT | N/A | N/A | White participants who conducted a virtual shooter game showed stronger implicit race bias against Black people when being embodied in a Black relative to White avatar. |
| Who is Credible (and Where)? Using Virtual Reality to Examine Credibility and Bias of Perceived Race/Ethnicity in Urban/Suburban Environments | Marino, M. I., Bilge, N., Gutsche, R. E., & Holt, L. | 2020 | USA | 248 | N/A | 66% Hispanic, 15% White, 14% Black, 3% Asian, 2% Native American or other | N/A | mixed | Interaction with an ingroup avatar | Hispanics in the US | 3D video | attitudes about the neighborhood; opinion of the information source (outgroup member); opinions of those who disliked the information source (outgroup member); attribution of a crime to the target outgroup member | N/A | N/A | Participants visited the scene of a break-in a 360˚ video, received a description of the robber from an actor, and then judged the source's credibility. Ratings of the source were more negative when the source was supposedly from negatively evaluated neighbourhoods. |
| Proxemic behaviors as predictors of aggression towards Black (but not White) males in an immersive virtual environment | McCall, C., Blascovich, J., Young, A., & Persky, S. | 2009 | USA | 47 | 26 | 68% White, 17% Hispanic, 8.5% Asian, 2.1% multiracial, 4.2% no identification | N/A | between-subjects | Interaction with an ingroup avatar | Black people | virtual agent | distance kept from target; shooting game; feelings towards virtual agents | N/A | N/A | Participants' proxemic behaviours (interpersonal distance and head movements) towards virtual agents were predictive of later shooting behaviour against these agents only for Black but not for White agents. |

(*Continued*)

**Table 1.** (Continued)

| Title | Authors | Year | Country of data collection | Total N | Percent female | Ethnic composition | Sample age | Type of design | Control condition | Prejudice target | Kind of interaction | Prejudice-related measures | Examined mediators | Examined moderators | Main results |
|---|---|---|---|---|---|---|---|---|---|---|---|---|---|---|---|
| Through Pink and Blue Glasses: Designing a Dispositional Empathy Game Using Gender Stereotypes and Virtual Reality | Müller, D. A., Van Kessel, C. R., & Janssen, S. | 2017 | Netherlands | 19 | N/A | N/A | N/A | between-subjects | N/A | Gender | virtual agents | empathy; willingness to act against sexism | N/A | N/A | Experiencing a virtual simulation of sex-discrimination situations from the point of view of both men and women led participants to self-report more dispositional empathy and perspective-taking and willingness to act in future discrimination situations. |
| Virtually old: Embodied perspective taking and the reduction of ageism under threat. (Study 1) | Oh, S. Y., Bailenson, J., Weisz, E., & Zaki, J. | 2016 | USA | 148 | 64 | 43% White, 32% Asian, 11% Latino, 7% Black, 7% other | 21 | within-subjects | Perspective taking exercise | elderly people | embodiment | perceived threat; explicit ageism; self-other overlap; future communication intentions | N/A | N/A | When confronted with intergenerational threat, young adult participants reported more self-other overlap with older adults after a perspective-taking exercise, especially when this was supported with the experience of embodying an avatar representing an older person. |
| Virtually old: Embodied perspective taking and the reduction of ageism under threat. (Study 2) | Oh, S. Y., Bailenson, J., Weisz, E., & Zaki, J. | 2016 | USA | 84 | 55 | 54% White, 12% Asian, 8% Latino, 13% Black, 13% Other | N/A | between-subjects | Perspective taking exercise | elderly people | embodiment | perceived threat; self-other overlap; future communication intentions; affect misattribution procedure; empathic listening task | N/A | N/A | Being socially excluded by older adults in a ball toss game, young adult participants showed less self-other overlap and more implicit preference for young over older people and this could not be overcome by an empathy task nor by embodying an older adult's virtual avatar. |
| Evidence of Racial Bias Using Immersive Virtual Reality: Analysis of Head and Hand Motions During Shooting Decisions | Peck, T. C., Good, J. J., & Seitz, K. | 2021 | USA | 99 | 56 | 80 White, 9 Asian, 8 Hispanic, 3 Black, 1 multi-racial | 23.1 | Within-subjects | Interaction with ingroup avatars | Black people | virtual agent | shooter bias (accuracy, latency, motion paths, bias scores) | N/A | Socioeconomic Status | Performing a shooter task in immersive virtual reality, participants showed no racial or socioeconomic bias in terms of latency to shoot but head and hand motion analyses predicted participants' implicit racial bias. |

(*Continued*)

**Table 1.** (Continued)

| Title | Authors | Year | Country of data collection | Total N | Percent female | Ethnic composition | Sample age | Type of design | Control condition | Prejudice target | Kind of interaction | Prejudice-related measures | Examined mediators | Examined moderators | Main results |
|---|---|---|---|---|---|---|---|---|---|---|---|---|---|---|---|
| Putting yourself in the skin of a black avatar reduces implicit racial bias | Peck, T. C., Seinfeld, S., Aglioti, S. M., & Slater, M | 2013 | Spain | 60 | 100 | N/A | N/A | between-subjects | embodying an ingroup or "alien-skinned" avatar OR seeing a non-embodied black-skinned virtual body | Black people | embodiment | race IAT | N/A | N/A | White participants who embodied a Black avatar reduced their implicit bias towards Black people more than participants embodying avatars with white or purple skin. |
| Virtual Reality and Political Outgroup Contact: Can Avatar Customization and Common Ingroup Identity Reduce Social Distance? | Peña, J, Wolff, G, Wojcieszak, M. | 2021 | USA | 149 | 100 | 38.2% Asian, 27.2% Latino, 29.9% Caucasian, 2.3% African American, 2.4% other | 20.3 (sd = 2.04) | between | embodying an avatar that looked like someone else; priming common ingroup identity (based on gender) | political outgroup | virtual agent | self-reported social distance | female identity salience | N/A | A sample of liberal women were either embodied in an avatar they customized to resemble themselves or someone else, and had their female identity either primed or not. They all met in VR a virtual agent representing a conservative woman. While avatar customization had an influence on social distance towards the political opponent, common ingroup identity priming did not. Female identity salience was found not to mediate the latter relationship. |
| Medical student bias and care recommendations for an obese versus non-obese virtual patient | Persky, S., & Eccleston, C. P. | 2011 | USA | 76 | 57 | 59% White, 32% Asian, 14% Black, 3% Hispanic | 26.2 | between-subjects | Interaction with ingroup avatar | Obese people | virtual agent | negative stereotyping; belief about patients' health; perceptions of patients' adherence; perception of patients' responsibility; visual contact; clinical recommendations | N/A | N/A | Medical students interacting with an obese (vs non-obese) virtual patient displayed more stereotyping, perceived the patient's health to be worse, attributed more responsibility, anticipated less patient adherence, and made less visual contact with the patient but did their clinical recommendations were unaffected by weight status. |

(*Continued*)

**Table 1.** (Continued)

| Title | Authors | Year | Country of data collection | Total N | Percent female | Ethnic composition | Sample age | Type of design | Control condition | Prejudice target | Kind of interaction | Prejudice-related measures | Examined mediators | Examined moderators | Main results |
|---|---|---|---|---|---|---|---|---|---|---|---|---|---|---|---|
| An Investigation Into The Impact of Virtual Reality Character Presentation on Participants' Depression Stigma | Rednmond, D., Hennessey, E., O'Connor, C., Balint, K., Parsons, T.D., Rooney, B. | 2019 | N/A | 54 | 48 | N/A | 22.06 (sd = 5.61) | between | vignette not priming depression; virtual agent trying to make eye contact | people with depression | virtual agent | personal stigma and perceived stigma towards depression | N/A | N/A | Participants met a male virtual agent in a café after reading a vignette either representing depression or not, while the virtual agent was either seeking eye contact with the participant or avoiding it. Neither experimental condition affected participants' level of stigma towards people with depression. |
| Cultivating Empathy Through Virtual Reality: Advancing Conversations About Racism, Inequity, and Climate in Medicine | Roswell, R. O., Cogburn, C. D., Tocco, J., Martinez, J., Bangeranye, C., Bailenson, J. N., Wright, M., Mieres, J. H., & Smith, L. | 2020 | USA | 76 | N/A | N/A | N/A | Within-subjects | N/A | Black people | 3D video | empathy; emotions regarding discrimination; experiences (qualitative data) | N/A | N/A | Participants of professional development sessions for medical school and health system leaders, faculty, and staff experienced three racial discrimination scenarios from the perspective of a Black male. Afterwards they reported more empathy towards minorities and felt that the experience helped them understand the experiences of other people. |
| The impact of virtual reality on implicit racial bias and mock legal decisions | Salmanowitz, N. | 2018 | USA | 92 | 50 | 100% White | 28 | between-subjects | experiencing the scenario without a virtual body | Black people | embodiment | Race IAT: Explicit attitudes re: race and gender (SRS, MSS); decision in mock crime scenario; follow-up after 5 days: another mock crime scenario and questions re: severity, likelihood of reoffending, an ambiguous legal case and evaluated an severity of sentence | N/A | N/A | After embodying a Black avatar, participants showed lower implicit racial bias and evaluated an ambiguous legal case more conservatively compared to participants who were immersed in a virtual world but did not embody an avatar. |
| The Effects of Embodiment in Virtual Reality on Implicit Gender Bias | Schulze, S., Pence, T., Irvine, N., & Guinn, C. | 2019 | USA | 16 | 31 | 93.8 White, 6.2 Black | N/A | between-subjects | embodying a male avatar | Women | embodiment | IAT "women and leadership" | N/A | N/A | Male and female participants were embodied in male or female avatars in an environment associated with leadership. Across all conditions there was a trend towards increased implicit bias against women in leadership positions. |

(*Continued*)

**Table 1.** (Continued)

| Title | Authors | Year | Country of data collection | Total N | Percent female | Ethnic composition | Sample age | Type of design | Control condition | Prejudice target | Kind of interaction | Prejudice-related measures | Examined mediators | Examined moderators | Main results |
|---|---|---|---|---|---|---|---|---|---|---|---|---|---|---|---|
| Shooter Bias in Virtual Reality: The Effect of Avatar Race and Socioeconomic Status on Shooting Decisions | Seitz, K. R., Good, J. J., & Peck, T. C. | 2020 | USA | 50 | N/A | N/A | N/A | between-subjects | Interaction with ingroup members | Black people | virtual agent | shooter bias | N/A | N/A | Participants in an immersive virtual shooter bias paradigm made fewer errors in trials with Black compared to White targets and were faster to shoot at agents that represented low, compared to high, socio-economic status and this was most pronounced for Black agents. |
| "I'm a Computer Scientist!": Virtual Reality Experience Influences Stereotype Threat and STEM Motivation Among Undergraduate Women | Starr, C. R., Anderson, B. R., & Green, K. A. | 2019 | USA | 79 | 100 | 46% Asian, 32% Latina, 14% White, 4% multiethnic | 20.3 | within-subjects | embodying a woman working in humanities | Women | embodiment | Self-STEM IAT; Gender-STEM IAT; Stereotype threat; expectancy beliefs; value beliefs | N/A | N/A | Female participants who embodied an avatar who had a successful career in science or technology showed (compared to a control condition that showed a career in humanities) higher course motivation, lower anticipated stereotype threat, stronger implicit associations between women and science/technology if they identified with the VR experience. |
| Can intergroup contact in virtual reality (Vr) reduce stigmatization against people with schizophrenia? | Stelzmann, D., Toth, R., & Schieferdecker, D. | 2021 | Germany | 114 | 58 | N/A | 24 (sd = 6.6) | between | watching the same video in 2D; other control condition (not specified) | schizophrenia patients | 3D video | anxiety, social proximity, empathy, and benevolence | N/A | N/A | Participants either watched a 3D or 2D video about a male schizophrenic patient describing life with schizophrenia, or were sorted in the control group. VR increased stigmatization of schizophrenic patients compared to the 2D condition, while it had no effects compared to the control condition. |

(*Continued*)

**Table 1.** (Continued)

| Title | Authors | Year | Country of data collection | Total N | Percent female | Ethnic composition | Sample age | Type of design | Control condition | Prejudice target | Kind of interaction | Prejudice-related measures | Examined mediators | Examined moderators | Main results |
|---|---|---|---|---|---|---|---|---|---|---|---|---|---|---|---|
| Body swapping with a Black person boosts empathy: Using virtual reality to embody another | Thériault, R., Olson, J.A., Krol, S.A., & Raz, A. | 2021 | Canada | 90 | 71 | 71% female, 29% male; 87% students; 50% White, 26% Asian, 17% South-Asian, 7% other | 22.2 (sd = 3.0) | between | mental perspective-taking exercise; no intergroup contact | Black people | embodiment | explicit measures, IAT, and empathy | N/A | N/A | Participants took the experiment with a Black person (confederate). Those in the experimental condition embodied said confederate's virtual body and saw through their perspective, while in the mental perspective-taking condition they had to imagine a day in the life of the confederate. No intergroup contact took place in the control condition. While the use of VR increased empathy compared to the control condition, there was no difference between the VR condition and the mental perspective-taking condition. |
| The design and evaluation of a body-sensing video game to foster empathy towards chronic pain patients | Tong, X., Ulas, S., Jin, W., Gromala, D., & Shaw, C. | 2017 | Canada | 15 | 27 | N/A | 24.8 | Within-subjects | N/A | Chronic pain patients | embodiment | Pommier Compassion Scale; Willingness to Help Scale | N/A | N/A | Participants showed stronger willingness to help with people living with chronic pain after embodying an avatar that simulated everyday situations from the point of view of people with chronic pain. |
| HIV-related stigma in social interactions: Approach and avoidance behaviour in a virtual environment | Toppenberg, H. L., Bos, A. E. R., Ruiter, R. A. C., Wigboldus, D. H. J., & Pryor, J. B. | 2015 | Netherlands | 50 | 74 | N/A | 22.3 | Within-subjects | Interaction with an ingroup avatar | HIV patients and homosexual men | virtual agent | Interpersonal distance; approach and walking away speed; time looking at virtual agent; homosexuality IAT; explicit attitudes towards homosexuals | N/A | N/A | When interacting with virtual agents in a virtual hospital setting, participants kept a larger interpersonal distance and approached the agent faster when the agent was depicted to have HIV rather than cancer, especially when the agent was depicted as being homosexual. HIV patients were looked at more often and less looked away from than other patients. Effects were unrelated to implicit and explicit attitudes. |

(*Continued*)

**Table 1.** (Continued)

| Title | Authors | Year | Country of data collection | Total N | Percent female | Ethnic composition | Sample age | Type of design | Control condition | Prejudice target | Kind of interaction | Prejudice-related measures | Examined mediators | Examined moderators | Main results |
|---|---|---|---|---|---|---|---|---|---|---|---|---|---|---|---|
| HIV status acknowledgment and stigma reduction in virtual reality: The moderating role of perceivers' attitudes. | Toppenberg, H. L., Ruiter, R. A. C., & Bos, A. E. R. | 2019 | Netherlands | 58 | 52 | N/A | 22.7 | within-subjects | N/A | HIV patients | virtual agent | HIV-IAT; explicit attitudes towards people with HIV | N/A | N/A | In a virtual job interview situation, participants evaluated HIV-positive virtual job applicants as more competent if they acknowledged their disease status and if their explicit attitudes towards people living with HIV were positive. Applications were evaluated more highly if they were not responsible for their infection. |
| The Effects of Immersive Virtual Reality in Reducing Public Stigma of Mental Illness in the University Population of Hong Kong: Randomized Controlled Trial | Yuen, A. S. Y. & Mak, W. W. S. | 2021 | Hong Kong | 206 | 55.3 | N/A | 21.76 (sd = 5.04, range: 18–64) | between/mixed | reading a written text from the perspective of a person with anxiety and depressive disorder; no intergroup contact condition | people with anxiety and depressive disorders | embodiment | public stigma towards people with mental illness (self-report) | sense of embodiment; story transportation | N/A | Embodying a person with anxiety and depressive disorders significantly reduced public stigma compared to the control condition, but not to the text condition. Both the embodied and text conditions have significant effects on public stigma over time (both at post-test and follow-up). Sense of embodiment and story transportation are sequential mediators of the effect of condition on public stigma. |
| Enfacing a female reduces the gender–science stereotype in males | Zhang, X., Hommel, B., & Ma, K. | 2021 | China | 97 | 100 | N/A | 21.49 (sd = 1.43, range: 19–25) | between/mixed | enfaced avatar with asynchronous movements | Women in science | embodiment | Gender-Science IAT | N/A | N/A | Participants enfaced in a female virtual avatar with synchronous movements showed decreased prejudice towards women in science after the experience compared to participants assigned to the control group (enfaced in female avatars with asynchronous movements). |

progress of research in the field. Of the remaining studies, 23 were published between 2015 and 2019, and 9 before 2014, the earliest publication being dated in 2007.

In terms of intergroup contact, in 28 of the included studies, participants are embodied in an avatar resembling an outgroup member, for example White participants being embodied in a Black avatar or male participants in a female avatar. 18 studies have participants interacting with a virtual agent, that is a virtual character steered by the computer, while only three studies use an avatar controlled by the experimenter. 13 studies present participants with 3D videos and in two studies, participants use augmented reality that enhances the perception of the real world with optical and auditory hallucinations as experienced by people living with schizophrenia.

The most commonly studied outcomes are explicit ($k = 37$) and implicit attitudes ($k = 25$) towards the target group with some form of the Implicit Association Test [11] being the most commonly used measure for the latter. Following the narrative of VR being an "empathy machine" [46,47], many studies include measures of empathy, sympathy, pity, self-other overlap, or willingness to help in the future ($k = 19$ studies using at least one such measure). Neurophysiological measures like heart rate, skin conductance, electroencephalography (EEG) or functional magnetic resonance imaging (fMRI) are applied much more rarely to examine physiological or neural activation patterns of prejudice during the experiments *($k = 5$)*. It is noteworthy that across studies, a wide range of different questionnaires, tasks, and measures is administered with seemingly no emerging standards in the field.

Lastly, 30 out of 64 studies control for the successfulness of the VR experience in terms of immersion, body ownership, or spatial presence, while the remaining ones failed to measure any related variable.

## 4.2. The effect of intergroup contact in VR on intergroup attitudes

The following paragraphs describe results from the included studies with a specific focus on the kind of contact experience created in VR, the kind of stigma that the outgroups represented, the outcome measures used to examine prejudice and intergroup bias, and the psychological mechanisms and moderating variables that have been studied. In our presentation of results, we place a special emphasis on key studies that we consider as positive examples in terms of rigorous methods and the results of which appear more trustworthy.

**4.2.1. Types of contact.** Two major forms of contact emerge from the analysed 64 studies: from the ingroup perspective, which is to say that participants belonging to a majority group interact in VR with avatars or virtual agents representing a stigmatised outgroup *($k = 28$)*; and from the outgroup perspective, meaning that the participant belonging to the majority group lives the virtual experience from a minority outgroup member's perspective ($k = 36$). In both types of designs, the subject does not always steer a virtual body: in some cases ($k = 13$), participants experience contact from a non-embodied virtual perspective using 360˚ videos, like for example in studies by Hasson et al. [48] and Lesur et al. [49]. Given that experiencing contact from the outgroup perspective is unique to VR, embodiment is by far the preferred method in the designs from a minority perspective (aside from a few exceptions, e.g. [49]), while those using a majority perspective are more often relying on a disembodied point of view (e.g. [50,51]). For a more precise overview of the methods chosen by each study, see Table 1.

When it comes to studies aiming at assessing prejudice in VR, only 3 out of 16 exploit the point of view of a minority group member. All of them seem to suggest that embodying an outgroup (minority group) member leads to positive outcomes such as increased empathy [52,53] and less implicit [53,54] and explicit [53] prejudice. Of the remaining 13 using the majority group perspective, most show that real-world prejudice can also be demonstrated in VR. For example, a pioneering randomised controlled trial by Dotsch and Wigboldus [55] has used

behavioural (the distance participants kept from avatars) and physiological (skin conductance responses) measures to show that participants exhibited higher prejudice towards Moroccan virtual agents rather than White ones.

In studies that adopt the majority perspective approach to prejudice (k = 28), both implicit and explicit measures of intergroup attitudes have been used providing somewhat inconsistent results. While some evidence [56,57] seems to suggest a decrease in prejudice towards minority outgroups following intergroup contact in VR, others [58–60] fail to obtain any significant change.

The same controversy appears when looking at the intervention studies carried out from a majority perspective, which mostly show a solid persistence of prejudice towards stigmatised minorities [51,55,61–69] Two recent studies [70,71] even report increased prejudice towards the contacted outgroup using explicit measures.

Among the intervention studies aiming at prejudice reduction through the majority perspective, two randomised controlled trials emerge [59,71]. While they both resort to explicit measures only, they have discording findings in that Kuuluvainen and colleagues [59] fail to find any improvement in intergroup anxiety after exposing White participants to virtual intergroup contact with a Middle Eastern man, compared to exposure to the same material in 2D. On the other hand, Peña et al. [71] have shown that participants who contacted a virtual outgroup member while embodying an avatar that resembled themselves, reported increased social distance towards the contacted political outgroup.

The results of the studies that adopt a minority perspective as a strategy to reduce prejudice (k = 36) are similarly mixed. For example, a randomised controlled trial exclusively targeting police officers [72] analysed participants' behavioural responses after being embodied in a Black suspect that was abused by another police officer, and eventually found greater helping behaviour up to one month after the VR experience. Banakou et al. [73] also show that experiencing the world from the minority outgroup's (i.e. Black people) perspective improves attitudes towards that group, and so do Peck et al. [74], Salmanowitz [75], Christofi et al. [76], Chen et al. [77,78], Chowdhury et al. [79], Tong et al. [53,80], and Zhang et al. [81]. However, other studies contest these findings by showing that intergroup contact experienced from the minority perspective does not necessarily have any effect on intergroup attitudes. Lastly, some scholars have found that impersonating an outgroup member may even worsen intergroup attitudes [82–86].

While more investigation is needed to understand the underlying mechanisms leading to increased prejudice in VR following the embodiment in an outgroup member, there is initial evidence suggesting that experiencing unpleasant circumstances in the skin of an outgroup member may lead to worsened attitudes rather than improved perspective taking towards the stigmatised outgroup. In the reviewed set of studies this was the case when participants experienced the point of view of people affected by Asperger syndrome [83], experienced schizophrenia symptoms simulated by augmented reality [84], or when White participants were embodied in a Black avatar [82]. To further validate this hypothesis, Banakou et al. [82] suggest that when participants experience negative affect while embodying outgroup members, their implicit bias against that group increases. The negative affect condition in this methodologically sound study was implemented as virtual passers-by displaying negative facial expressions, staring right at the participant, and changing direction to avoid participants. Whereas Kishore et al. [72] reach opposite conclusions, it is worth pointing out that their trial had a significantly smaller sample size, which makes those findings less reliable. We will next examine potential target-specific effects of VR contact more thoroughly, and then move to mediators and moderators of the effect of VR contact on prejudice in section 4.2.4.

**4.2.2 Types of stigma.** Of the 64 articles found eligible for inclusion, 31 focus on tribal stigma, with most studies focusing on contact with avatars representing outgroups of African

ethnic background. 13 studies deal with deviations in personal traits (e.g. schizophrenia, HIV, substance abuse), and 8 with stigma deriving from overt or external deformations. Of the latter, 4 target elderly people, 2 people with physical disabilities, and 2 individuals with obesity. Finally, among the 12 studies targeting intersectional types of stigma, one tackles prejudice towards transgender people, and the remaining ones towards women.

Different correlational designs exploring intergroup bias [55,61,87] confirm persisting bias against people with African background in VR, including studies that show persistence of the "shooter bias" (i.e. that participants tend to shoot more often and faster at Black rather than White targets in ambiguous shooting situations, [88]) in VR [62,66,68].

However, there is also evidence that both intergroup contact [56,72] and embodiment in an outgroup avatar [74,77,78] in VR can successfully be used to decrease racial bias. Furthermore, evidence by Hasler et al. [56] and Hasson et al. [48] suggests that the effect of VR contact is not specific for interracial attitudes, but can also improve in other, critical intergroup conflict situations: both studies showed that Jewish Israelis' attitude towards Palestinians could be improved using VR. While participants in Hasler et al.'s [56] study achieved this through a discussion with an outgroup avatar, Hasson and colleagues [48] obtained positive results using 3D videos to present the outgroup's perspective.

While positive VR interaction has shown its potential in reduction of racial prejudice, as already discussed above, there is some contrasting evidence of the effect of embodiment in a racial minority group member in VR on intergroup attitudes. Namely, there are results showing that embodying Black avatars can also lead to worsened implicit attitudes in White participants [89]. The previously mentioned study by Banakou et al. [82] shows that negative contact conditions when embodying a Black avatar and the associated affective reaction can be one explanatory factor for this effect. On the other hand, Kishore et al. [72] find that being embodied in a Black avatar targeted by discriminating behaviour leads to greater helping behaviour. Finally, two trials using either embodiment in an outgroup avatar [90] or 3D videos of an interaction with a Middle Eastern man [59], fail to find any improvement in intergroup attitudes compared to the control group.

When it comes to deviations in personal traits, Toppenberg et al. [51,69] show that implicit bias towards people living with HIV persists even in VR, and that evaluations were more positive when they perceived responsibility for the condition was low. While Tong et al. [53,80] take it a step further, proving that being embodied in chronic pain patients improve self-reported attitudes and willingness to help, contrasting evidence is brought by designs using augmented reality to simulate schizophrenia symptoms. Interestingly, while de Silva et al. [91] show increased empathy towards schizophrenic patients following an augmented reality experience, Kalyanaraman et al. [84] suggest that such embodied experience may lead to a desire for keeping a greater distance towards them. Stelzmann et al. [70] also find stronger stigmatisation of people with schizophrenia after facing an outgroup member in a 3D video. Hadjipanayi and Michel-Grigoriou [83] reach similar conclusions following embodiment in people with Asperger syndrome. Interestingly, Peña et al. [71] suggest that embodying an avatar that physically resembles the self leads to increased social distance towards a contacted political outgroup. Finally, Yuen et al. [92] fail to find any difference between VR embodiment in a subject with depressive symptoms compared to text-based perspective-taking.

As far as stigma due to overt or external deformations is concerned, Persky and Eccleston [67] show that obese virtual patients are object to prejudiced treatment when dealing with health professionals. Chowdhury et al. [79] find a decrease in prejudice towards wheelchair users following virtual embodiment. Moreover, a contrasting trend is shown by Banakou et al. [93], who have found embodiment in an elderly individual with high IQ improves implicit

attitudes toward elderly people, and Oh et al. [94], whose subjects did not show any improvement in attitudes after being embodied in an elderly woman.

Lastly, designs dealing with intersectional stigma highlight the same pattern of mixed evidence when it comes to interventions. Indeed, while some fail to find any positive effect of intergroup VR contact [49] and embodiment in an outgroup member [95], others show improved attitudes can be a result of both methods [57,81]. On the other hand, two studies [85,86] suggest that embodying male individuals in female avatars may also lead to the deterioration of implicit attitudes, even when the performed task is not supposed to elicit any negative affect (i.e. a Tai-Chi class). Observational studies on intersectional stigma confirmed the endurance of gender-based bias [63,64], and bias based on sexual orientation [96].

The results above reinforce the hypothesis that factors such as the degree of immersion and the valence (positive or pleasant vs negative or unpleasant) of the embodied experience may play a primary role in the success or failure to reduce prejudice following embodiment in an outgroup member.

**4.2.3. Outcome measure.** Given the previously discussed and widely studied differences between explicit and implicit measures of prejudice, it is worth discussing the findings also on the basis of their outcome measures. We chose not to focus on results obtained through physiological and neurological measures, due to them being severely underrepresented in the included studies. Specifically, only 5 of the included studies include a neurophysiological measure of prejudice, as compared to 37 that assess explicit, and 25 implicit attitudes. Of those, 10 assess prejudice with both implicit and explicit measures.

Twelve out of 25 studies examining implicit intergroup attitudes represent intervention studies and rely on IAT to assess intergroup bias. Of those, Lopez et al. [85] and Schulze et al. [86] found that implicit attitudes further deteriorated after the intervention, while Banakou et al. [73,93], Peck et al. [74], Starr et al. [54], and Zhang et al. [81] highlight a clear improvement in implicit attitudes following exposure to VR contact. Notably, all intervention studies assessing implicit attitudes are based on embodiment of an outgroup member.

By contrast, sixteen studies enacting bias-reducing interventions exclusively used explicit measures. A considerable number of them found a decrease in prejudice following embodiment in an outgroup member [52,53,76–78,80,92]. Two studies by Peña et al. [71] and Steltzmann et al. [70] conversely found increased levels of prejudice after engaging in virtual intergroup contact with an outgroup member, and Hadjipanayi and Michel-Grigoriou [83] and Kalyanaraman et al. [84] obtain similar results through embodiment of an outgroup member.

Lastly, sixteen intervention studies include both implicit and explicit measures of prejudice, of which nine focus on a majority perspective. Whereas three of them report a decrease in intergroup bias assessed through implicit measures after embodiment of an outgroup member, but no significant change in explicit ones [75,82,97], Breves [40] only found a decrease in prejudice through explicit measures, but no effect on implicit ones. Finally, Groom et al. [89] show that embodying an outgroup member in a work interview leads to worse implicit attitudes but has no effect on explicit ones. No intervention study taking into consideration both implicit and explicit measures of prejudice has found converging results, but as already pointed out earlier, these two types of measurements are often discordant, most likely due to social desirability effects (for an overview of the discussion about the discordance between implicit and explicit measures, see e.g., [13,98–101]). In addition, most studies employing implicit measures used embodiment of an outgroup member as an intervention which might be more likely to change implicit rather than explicit attitudes. In summary, it seems like implicit measures unveil potential effects of VR-based interventions that might not appear in explicit measures of intergroup attitudes.

**4.2.4. Mediators and moderators.** It is widely established that intergroup contact reduces prejudice through both affective (i.e. empathy, intergroup anxiety) and cognitive mediators (i.e. perspective taking, increased familiarity, and knowledge; see [25] for a review). Fourteen studies encompass an analysis of potential mediating mechanisms explaining the effect of VR contact on prejudice. Among those, six are observational studies. Two of them suggest that physiological measures have great potential to elucidate mechanisms accounting for prejudice in VR. Specifically, [61] have found an association between EEG-measured alertness and attitudes, while Dotsch and Wigboldus [55] have observed that measures of skin conductance are correlated to implicit attitudes towards the target minority. On the other hand, regarding potential psychological mediators, Eiler [62] have found no mediation of perceived threat on prejudiced behaviour, nor Bielen et al. [87] of concern about terrorism when judging minority defendants in a court trial.

When it comes to prejudice-reducing interventions, evidence emerges that the positive effect of VR contact is due to emotional mediators, such as feeling more closeness to the prejudiced target [76] and perceiving them as warmer [58]. Empathy has also been found to be a mediator of VR contact when it comes to embodying an outgroup member [77]. Furthermore, Hasler et al. [102] interestingly show that feelings of presence in VR had a mediating effect on the negative affect toward the majority ingroup, when experiencing a conflict scenario from the outgroup's (minority) perspective. Lastly, Peña et al. [71] showed that inducing identity salience does not mediate changes in prejudice.

Thirteen studies include moderation analyses. Among the ones delivering interventions, few studies investigated individual differences as moderating variables. Christofi et al. [76] have found that differences in trait empathy moderate the improvement of attitudes towards the outgroup, with individuals higher in empathy showing less bias after VR contact than those low in empathy. Additionally, two studies investigated social identification as a moderating variable on the effects of embodying an outgroup avatar. Chen et al. [77] show that participants generally placing greater importance on their various group memberships show stronger intervention effects, namely greater increase in self-other overlap with the embodied ethnic outgroup. Starr et al. [54] suggest that higher identifiers with the embodied avatar experience greater decrease in intergroup bias.

When it comes to moderators linked to specific features of the VR experience, Chowdhury et al. [79] interestingly found that a disabled narrator led to greater decrease in prejudice against disabled people, when embodying a wheelchair user. In addition, Banakou et al. [82] show that the valence of intergroup contact while embodying an outgroup member moderates the change in attitudes towards the embodied minority (with more positive contact resulting in a more positive change), while the number of exposures to the same kind of embodiment-based intervention does not [73]. Finally, Peña et al. [71] found that participants customising their own avatar to look like themselves eventually expressed desire for greater social distance from the contacted outgroup following the interaction.

The observational studies using the shooter bias paradigm revealed no effect of distance and armed status on difference of shooting behaviour towards majority or minority members [62], but a moderation effect of socioeconomic status (SES) on shooting accuracy [103], with subjects making fewer mistakes when facing high-SES targets.

## 4.3 Risk of bias assessment

The risk of bias assessment (see Table 2 for the detailed risk of bias assessment and S1 Table in S1 Checklist for the overview) showed that a large proportion of studies did not report specifically how participants were assigned to conditions. Relatedly, it was also often unclear to what

**Table 2. Risk of bias assessment for all studies following the Cochrane Collaboration's risk of bias tool (Higgins et al, 2019).** As detection bias is assessed for each outcome separately, we classified the different outcomes when more than one was reported in a study and rated the risk of bias for each class of outcomes (e.g. IAT; self-reports; physiological recordings) as low, unclear, or high.

| Title | Authors | Selection bias: random sequence generation | Selection bias: allocation concealment | Performance bias: experimenter | Performance bias: participant | Detection bias (rated for each outcome) | | | | | | Attrition bias (participants) | Attrition bias (outcome) | Reporting bias |
|---|---|---|---|---|---|---|---|---|---|---|---|---|---|---|
| | | | | | | rated outcome | rating | rated outcome | rating | rated outcome | rating | | | |
| Contact in VR: Testing Avatar Customisation and Common Ingroup Identity Cues on Outgroup Bias Reduction | Alvidrez, S. & Peña, J. | unclear | unclear | high | low | self-report | unclear | -- | -- | -- | -- | unclear | unclear | low |
| Verbal Mimicry Predicts Social Distance and Social Attraction to an Outgroup Member in Virtual Reality | Alvidrez, S. & Peña, J. | unclear | unclear | high | low | self-report | unclear | verbal distance | low | -- | -- | unclear | unclear | low |
| Virtual body ownership and its consequences for implicit racial bias are dependent on social context | Banakou, D., Beacco, A., Neyret, S., Blasco-Oliver, M., Seinfeld, S., & Slater, M. | high | unclear | unclear | unclear | IAT | low | self-reported prejudice | high | -- | -- | unclear | unclear | low |
| Virtual Embodiment of White People in a Black Virtual Body Leads to a Sustained Reduction in Their Implicit Racial Bias | Banakou, D., Hanumanthu, P. D., & Slater, M. | unclear | unclear | unclear | unclear | IAT | low | self-reported prejudice | high | -- | -- | unclear | unclear | low |
| Virtually Being Einstein Results in an Improvement in Cognitive Task Performance and a Decrease in Age Bias | Banakou, D., Kishore, S., & Slater, M. | unclear | unclear | unclear | unclear | IAT | low | -- | -- | -- | -- | unclear | unclear | low |
| Racial bias and in-group bias in virtual reality courtrooms. | Bielen, S., Marneffe, W., & Mocan, N. | unclear | unclear | unclear | unclear | judgments | unclear | concern about terrorism | unclear | -- | -- | unclear | unclear | unclear |
| Presence, what is is good for? Exploring the benefits of virtual reality at evoking empathy towards the marginalized | Boehm, N. | unclear | unclear | high | unclear | self-reports | unclear | -- | -- | -- | -- | low | unclear | low |

(*Continued*)

**Table 2.** (Continued)

| Title | Authors | Selection bias: random sequence generation | Selection bias: allocation concealment | Performance bias: experimenter | Performance bias: participant | rated outcome | rating | Detection bias (rated for each outcome) | | | | | Attrition bias (participants) | Attrition bias (outcome) | Reporting bias |
|---|---|---|---|---|---|---|---|---|---|---|---|---|---|---|---|
| | | | | | | | | rated outcome | rating | rated outcome | rating | rated outcome | | | |
| Reducing Outgroup Bias through Intergroup Contact with Non-Playable Video Game Characters in VR | Breves, P. | unclear | low | high | low | IAT | low | -- | -- | -- | -- | -- | unclear | unclear | low |
| Perspective-Taking in Virtual Reality and Reduction of Biases against Minorities (Study 1) | Chen, V., Chan, S. & Tan, Y. | unclear | unclear | unclear | unclear | self-reports | unclear | -- | -- | -- | -- | -- | unclear | unclear | low |
| The Effect of VR Avatar Embodiment on Improving Attitudes and Closeness Toward Immigrants | Chen, V., Ibasco, G., Leow, V., & Lew, J. | unclear | unclear | unclear | unclear | self-reports | unclear | -- | -- | -- | -- | -- | low | low | unclear |
| A Virtual Reality Simulation of Drug Users' Everyday Life: The Effect of Supported Sensorimotor Contingencies on Empathy | Christofi, M., Michael-Grigoriou, D., & Kyrlitsias, C. | unclear | high | high | high | self-reports | unclear | -- | -- | -- | -- | -- | low | low | low |
| VR Disability Simulation Reduces Implicit Bias Towards Persons With Disabilities | Chowdhury, T., Ferdous, S., & Quarles, J. | unclear | unclear | high | unclear | IAT | low | -- | -- | -- | -- | -- | low | unclear | unclear |
| A Wheelchair Locomotion Interface in a VR Disability Simulation Reduces Implicit Bias | Chowdhury, T. & Quarles, J. | unclear | unclear | high | unclear | IAT | low | -- | -- | -- | -- | -- | low | unclear | unclear |
| Influence of weight etiology information and trainee characteristics on Physician-trainees' clinical and interpersonal communication. | Cohen, R. W., & Persky, S. | unclear | unclear | unclear | unclear | communication outcomes | low | -- | -- | -- | -- | -- | low | low | low |

(Continued)

**Table 2.** (Continued)

| Title | Authors | Selection bias: random sequence generation | Selection bias: allocation concealment | Performance bias: experimenter | Performance bias: participant | rated outcome | Detection bias (rated for each outcome) rating | rated outcome | rating | rated outcome | rating | Attrition bias (participants) | Attrition bias (outcome) | Reporting bias |
|---|---|---|---|---|---|---|---|---|---|---|---|---|---|---|
| Using virtual reality to induce gratitude through virtual social interaction | Collange, J., & Guegan, J. | unclear | high | high | low | self-reports | unclear | -- | -- | -- | -- | low | low | low |
| Prosocial Virtual Reality, Empathy, and EEG Measures: A Pilot Study Aimed at Monitoring Emotional Processes in Intergroup Helping Behaviors | D'Errico, F., Leone, G., Schmid, M., & D'Anna, C. | unclear | unclear | unclear | unclear | EEG measures | low | -- | -- | -- | -- | low | low | low |
| Reducing the schizophrenia stigma: A new approach based on augmented reality. | de C. Silva, R. D., Albuquerque, S. G. C., de V. Muniz, A., Reboucas Filho, P. P., Ribeiro, S., Pinheiro, P. R., & Albuquerque, V. H. C. | high | high | N/A | N/A | self-reports | high | -- | -- | -- | -- | low | low | low |
| Virtual prejudice | Dotsch, R., & Wigboldus, D. H. J. | unclear | low | low | unclear | distance from avatar in VR | low | skin conductance | low | IAT | low | low | low | low |
| The behavioral dynamics of shooter bias in virtual reality: The role of race, armed status, and distance on threat perception and shooting dynamics | Eiler, B. A. | low | low | low | unclear | shooter bias | low | -- | -- | -- | -- | low | low | low |
| Virtual Virgins and Vamps: The Effects of Exposure to Female Characters' Sexualized Appearance and Gaze in an Immersive Virtual Environment | Fox, J., & Bailenson, J. | unclear | unclear | unclear | unclear | self-report | unclear | -- | -- | -- | -- | low | low | unclear |

(*Continued*)

**Table 2.** (Continued)

| Title | Authors | Selection bias: random sequence generation | Selection bias: allocation concealment | Performance bias: experimenter | Performance bias: participant | Detection bias (rated for each outcome) | | | | | | Attrition bias (participants) | Attrition bias (outcome) | Reporting bias |
|---|---|---|---|---|---|---|---|---|---|---|---|---|---|---|
| | | | | | | rated outcome | rating | rated outcome | rating | rated outcome | rating | | | |
| The Effect of Embodying a Woman Scientist in Virtual Reality on Men's Gender Biases | Freedman, G., Green, M.C., Seidman, M., & Flanagan, M. | low | low | unclear | low | explicit attitudes | unclear | IAT | low | -- | -- | low | low | low |
| Psychological response to an emergency in virtual reality: Effects of victim ethnicity and emergency type on helping behavior and navigation | Gamberini, L., Chittaro, L., Spagnolli, A., & Carlesso, C. | unclear | unclear | unclear | unclear | discrimination | low | -- | -- | -- | -- | low | low | low |
| Being the Victim of Intimate Partner Violence in Virtual Reality: First- Versus Third-Person Perspective | Gonzalez-Liencres, C., Zapata, L. E., Iruretagoyena, G., Seinfeld, S., Perez-Mendez, L., Arroyo-Palacios, J., Borland, D., Slater, M., & Sanchez-Vives, M. V | unclear | high | high | unclear | physiological measures | low | IAT | low | -- | -- | low | low | low |
| The influence of racial embodiment on racial bias in immersive virtual environments | Groom, V., Bailenson, J. N., & Nass, C. | unclear | high | high | low | IAT | low | Interpersonal distance | low | MRS and RAS | unclear | low | low | low |
| Virtual Humans and Persuasion: The Effects of Agency and Behavioral Realism | Guadagno, R. E., Blascovich, J., Bailenson, J. N., & McCall, C. | unclear | unclear | unclear | unclear | agreement with agent's argument | unclear | impression of virtual agent | low | -- | -- | low | low | low |
| Conceptual knowledge and sensitization on Asperger's syndrome based on the constructivist approach through virtual reality | Hadjipanayi, C., & Michael-Grigoriou, D. | unclear | high | high | unclear | sensitization | unclear | -- | -- | | | low | low | low |
| Virtual race transformation reverses racial ingroup bias | Hasler, B. S., Spanlang, B., & Slater, M. | unclear | unclear | unclear | unclear | IAT | low | mimicry | low | | | low | low | low |

(Continued)

**Table 2.** (Continued)

| Title | Authors | Selection bias: random sequence generation | Selection bias: allocation concealment | Performance bias: experimenter | Performance bias: participant | rated outcome | Detection bias (rated for each outcome) | | | | | Attrition bias (participants) | Attrition bias (outcome) | Reporting bias |
|---|---|---|---|---|---|---|---|---|---|---|---|---|---|---|
| | | | | | | | rating | rated outcome | rating | rated outcome | rating | | | |
| Virtual Peacemakers: Mimicry Increases Empathy in Simulated Contact with Virtual Outgroup Members | Hasler, B. S., Hirschberger, G., Shani-Sherman, T., & Friedman, D. A. | unclear | high | high | unclear | empathy | unclear | self-other overlap | unclear | outgroup affect | unclear | high | low | low |
| Virtual Reality-based Conflict Resolution: The Impact of Immersive 360° Video on | Hasler, B., Hasson, Y., Landau, D., Eyal, N. S., Giron, J., Levy, J., Halperin, E., & Friedman, D | unclear | unclear | unclear | unclear | self-reports | unclear | physiological measurements | low | -- | -- | low | low | unclear |
| The enemy's gaze: Immersive virtual environments enhance peace promoting attitudes and emotions in violent intergroup conflicts (Study 1) | Hasson, Y., Schori-Eyal, N., Landau, D., Hasler, B. S., Levy, J., Friedman, D., & Halperin, E. | unclear | unclear | unclear | unclear | self-reports | unclear | -- | -- | -- | -- | low | low | low |
| The enemy's gaze: Immersive virtual environments enhance peace promoting attitudes and emotions in violent intergroup conflicts (Study 2) | Hasson, Y., Schori-Eyal, N., Landau, D., Hasler, B. S., Levy, J., Friedman, D., & Halperin, E. | unclear | unclear | unclear | unclear | self-reports | unclear | -- | -- | -- | -- | low | low | low |
| The effect of gender, religiosity and personality on the interpersonal distance preference: a virtual reality study | Hatami, J., Sharifian, M., Noorollahi, Z., & Fathipour, A. | low | low | low | unclear | preferred distance | unclear | -- | -- | -- | -- | unclear | low | low |
| The Virtual Doppelganger Effects of a Virtual Reality Simulator on Perceptions of Schizophrenia | Kalyanaraman, S. S., Penn, D. L., Ivory, J. D., & Judge, A. | unclear | high | high | unclear | self-reports | unclear | -- | -- | -- | -- | low | low | low |

(*Continued*)

**Table 2.** (Continued)

| Title | Authors | Selection bias: random sequence generation | Selection bias: allocation concealment | Performance bias: experimenter | Performance bias: participant | rated outcome | Detection bias (rated for each outcome) | | | | | Attrition bias (participants) | Attrition bias (outcome) | Reporting bias |
|---|---|---|---|---|---|---|---|---|---|---|---|---|---|---|
| | | | | | | | rating | rated outcome | rating | rated outcome | rating | | | |
| Processing Racial Stereotypes in Virtual Reality: An Exploratory Study Using Functional Near-Infrared Spectroscopy (fNIRS) | Kim, G., Buntain, N., Hirshfield, L., Costa, M. R., & Chock, T. M. | unclear | high | unclear | unclear | brain activation | low | -- | -- | -- | -- | low | low | low |
| A Virtual Reality Embodiment Technique to Enhance Helping Behavior of Police Towards a Victim of Police Racial Aggression | Kishore, S., Spanlang, B., Iruretagoyena, G., Halan, S., Szostak, D., & Slater, M. | low | unclear | low | unclear | helping behavior | low | -- | -- | -- | -- | low | low | low |
| Testing an Immersive Virtual Environment for Decreasing Intergroup Anxiety among University Students: An Interpersonal Perspective | Kuuluvainen, V., Virtanen, I., Rikkonen, L., & Isotalus, P. | low | unclear | high | low | intergroup anxiety | high | -- | -- | -- | -- | low | low | low |
| No Country for Old Men: Reducing Age Bias through Virtual Reality Embodiment | La Rocca, S., Brighenti, A., Tosi, G., & Daini, R. | unclear | low | low | unclear | IAT | low | Ageism Scale | unclear | -- | -- | low | unclear | low |
| How Does Embodying a Transgender Narrative Influence Social Bias? An Explorative Study in an Artistic Context | Lesur, M. R., Lyn, S., & Lenggenhager, B. | high | high | high | low | IAT | low | Explicit attitudes towards transgender | unclear | -- | -- | low | low | low |
| Humans adjust virtual comfort-distance towards an artificial agent depending on their sexual orientation and implicit prejudice against gay men | Lisi, M.P., Fusaro, M., Tieri, G., & Aglioti, S.M. | unclear | unclear | unclear | high | | low | | low | | high | low | low | low |

(*Continued*)

**Table 2.** (Continued)

| Title | Authors | Selection bias: random sequence generation | Selection bias: allocation concealment | Performance bias: experimenter | Performance bias: participant | rated outcome | Detection bias (rated for each outcome) | | | | | | Attrition bias (participants) | Attrition bias (outcome) | Reporting bias |
|---|---|---|---|---|---|---|---|---|---|---|---|---|---|---|---|
| | | | | | | | rating | rated outcome | rating | rated outcome | rating | | | |
| Investigating Implicit Gender Bias and Embodiment of White Males in Virtual Reality with Full Body Visuomotor Synchrony | Lopez, S., Yang, Y., Beltran, K., Kim, S. J., Hernandez, J. C., Simran, C., Yang, B., & Yuksel, B. F. | unclear | unclear | unclear | unclear | IAT | low | -- | -- | -- | -- | low | low | low |
| Mitigating Negative Effects of Immersive Virtual Avatars on Racial Bias | Maloney, D. | unclear | unclear | unclear | high | IAT | low | -- | -- | -- | -- | high | low | high |
| Who is Credible (and Where)? Using Virtual Reality to Examine Credibility and Bias of Perceived Race/Ethnicity in Urban/Suburban Environments | Marino, M. L., Bilge, N., Gutsche, R. E., & Holt, L. | high | high | high | unclear | self-reports | unclear | -- | -- | -- | -- | low | low | low |
| Proxemic behaviors as predictors of aggression towards Black (but not White) males in an immersive virtual environment | McCall, C., Blascovich, J., Young, A., & Persky, S. | unclear | unclear | unclear | unclear | proximity | low | shooting task data | low | feelings towards the agent | high | low | low | low |
| Through Pink and Blue Glasses: Designing a Dispositional Empathy Game Using Gender Stereotypes and Virtual Reality | Muller, D. A., Van Kessel, C. R., & Janssen, S. | high | high | unclear | unclear | self-reports | unclear | -- | -- | -- | -- | low | low | unclear |
| Virtually old: Embodied perspective taking and the reduction of ageism under threat. (Study 1) | Oh, S. Y., Bailenson, J., Weisz, E., & Zaki, J. | unclear | high | high | unclear | self-reports | unclear | -- | -- | -- | -- | low | low | low |
| Virtually old: Embodied perspective taking and the reduction of ageism under threat. (Study 2) | Oh, S. Y., Bailenson, J., Weisz, E., & Zaki, J. | unclear | high | high | unclear | self-reports | unclear | affect misattribution procedure | low | Empathic Listening Task | low | high | low | low |

*(Continued)*

**Table 2.** (Continued)

| Title | Authors | Selection bias: random sequence generation | Selection bias: allocation concealment | Performance bias: experimenter | Performance bias: participant | Detection bias (rated for each outcome) | | | | | | Attrition bias (participants) | Attrition bias (outcome) | Reporting bias |
|---|---|---|---|---|---|---|---|---|---|---|---|---|---|---|
| | | | | | | rated outcome | rating | rated outcome | rating | rated outcome | rating | | | |
| Putting yourself in the skin of a black avatar reduces implicit racial bias | Peck, T. C., Good, J. J., & Seitz, K. | unclear | unclear | unclear | unclear | IAT | low | -- | -- | -- | -- | low | low | low |
| Evidence of Racial Bias Using Immersive Virtual Reality: Analysis of Head and Hand Motions During Shooting Decisions | Peck, T. C., Seinfeld, S., Aglioti, S. M., & Slater, M | unclear | low | low | low | IAT | low | movement data etc | low | -- | -- | low | unclear | low |
| Virtual Reality and Political Outgroup Contact: Can Avatar Customization and Common Ingroup Identity Reduce Social Distance? | Peña, J., Wolff, G., & Wojcieszak, M. | low | unclear | high | low | social distance | high | -- | -- | -- | -- | low | low | unclear |
| Medical student bias and care recommendations for an obese versus non-obese virtual patient | Persky, S., & Eccleston, C. P. | unclear | unclear | unclear | unclear | self-reports | unclear | visual contact | low | -- | -- | low | low | low |
| An Investigation Into The Impact of Virtual Reality Character Presentation on Participants' Depression Stigma | Redmond, D., Hennessey, E., O'Connor, C., Balint, K., Parsons, T.D., & Rooney, B. | unclear | unclear | unclear | low | | low | -- | -- | -- | -- | low | low | high |
| Cultivating Empathy Through Virtual Reality: Advancing Conversations About Racism, Inequity, and Climate in Medicine | Roswell, R. O., Cogburn, C. D., Tocco, J., Martinez, J., Bangeranye, C., Bailenson, J. N., Wright, M., Mieres, J. H., & Smith, L. | high | high | high | high | self-reports | high | -- | -- | -- | -- | high | low | low |
| The impact of virtual reality on implicit racial bias and mock legal decisions | Salmanowitz, N. | unclear | unclear | unclear | low | mock legal decision | low | IAT | low | explicit attitudes | high | low | low | low |
| The Effects of Embodiment in Virtual Reality on Implicit Gender Bias | Schulze, S., Pence, T., Irvine, N., & Guinn, C. | unclear | unclear | unclear | unclear | IAT | low | -- | -- | -- | -- | low | low | low |

*(Continued)*

**Table 2.** (Continued)

| Title | Authors | Selection bias: random sequence generation | Selection bias: allocation concealment | Performance bias: experimenter | Performance bias: participant | rated outcome | Detection bias (rated for each outcome) | | | | | Attrition bias (participants) | Attrition bias (outcome) | Reporting bias |
|---|---|---|---|---|---|---|---|---|---|---|---|---|---|---|
| | | | | | | | rating | rated outcome | rating | rated outcome | rating | | | |
| Shooter Bias in Virtual Reality: The Effect of Avatar Race and Socioeconomic Status on Shooting Decisions | Seitz, K. R., Good, J. J., & Peck, T. C. | unclear | unclear | low | low | shooter bias | low | -- | -- | -- | -- | low | low | high |
| "I'm a Computer Scientist!": Virtual Reality Experience Influences Stereotype Threat and STEM Motivation Among Undergraduate Women | Starr, C. R., Anderson, B. R., & Green, K. A. | unclear | unclear | unclear | low | IATs | low | self-reports | unclear | -- | -- | low | low | low |
| Can intergroup contact in virtual reality (Vr) reduce stigmatization against people with schizophrenia? | Stelzmann, D., Toth, R., & Schieferdecker, D. | low | low | high | low | all self reports | high | -- | -- | -- | -- | low | low | low |
| Body swapping with a Black person boosts empathy: Using virtual reality to embody another | Thériault, R., Olson, J.A., Krol, S.A., & Raz, A. | low | high | high | unclear | IAT | low | self reported racial bias | high | empathy | unclear | low | low | low |
| Designing a Virtual Reality Game for Promoting Empathy Toward Patients With Chronic Pain: Feasibility and Usability Study. | Tong, X., Gromala, D., Kiaei Ziabari, S. P., & Shaw, C. D. | high | high | high | high | all self-report | high | -- | -- | -- | -- | low | low | low |
| The design and evaluation of a body-sensing video game to foster empathy towards chronic pain patients | Tong, X., Ulas, S., Jin, W., Gromala, D., & Shaw, C. | high | high | high | high | all self-report | high | -- | -- | -- | -- | low | low | low |

(*Continued*)

**Table 2.** (Continued)

| Title | Authors | Selection bias: random sequence generation | Selection bias: allocation concealment | Performance bias: experimenter | Performance bias: participant | rated outcome | rating | Detection bias (rated for each outcome) | | | | | | Attrition bias (participants) | Attrition bias (outcome) | Reporting bias |
|---|---|---|---|---|---|---|---|---|---|---|---|---|---|---|---|---|
| | | | | | | | | rated outcome | rating | rated outcome | rating | rated outcome | rating | | | |
| HIV-related stigma in social interactions: Approach and avoidance behaviour in a virtual environment | Toppenberg, H. L., Bos, A. E. R., Ruiter, R. A. C., Wigboldus, D. H. J., & Pryor, J. B. | unclear | unclear | low | high | distance, speed, head orientation | low | IAT | low | explicit attitudes | high | | | low | low | low |
| HIV status acknowledgment and stigma reduction in virtual reality: The moderating role of perceivers' attitudes. | Toppenberg, H. L., Ruiter, R. A. C., & Bos, A. E. R. | unclear | unclear | low | high | IAT | low | explicit attitudes | high | evaluation of job candidate | unclear | | | low | low | low |
| The Effects of Immersive Virtual Reality in Reducing Public Stigma of Mental Illness in the University Population of Hong Kong: Randomized Controlled Trial | Yuen, A. S. Y. & Mak, W. W. S. | low | unclear | high | low | Perception of public stigma | low | -- | -- | -- | -- | | | low | low | low |
| Enfacing a female reduces the gender–science stereotype in males | Zhang, X., Hommel, B. &, Ma, K. | low | unclear | unclear | unclear | IAT | low | -- | -- | -- | -- | | | low | low | low |

degree experimenters and participants were aware of which condition participants were assigned to, inducing risk for performance biases in both participants and experimenters.

Judging from the written reports, risk for performance bias in participants was deemed to be high or unclear in over 80% of studies. This was mostly because it was not clearly reported whether participants could have guessed the purpose of the study and/or which condition they were assigned to.

The overall low risk of reporting and attrition bias is worth positive mention: most studies reported null results for at least some of the assessed variables. However, without pre-registration, it is impossible to assess whether further variables were assessed but not reported.

## 5. Discussion

First and foremost, this systematic review shows that VR is not a social vacuum but a virtual environment enabling co-creation and modification of social reality. The review thus clearly indicates that features of prejudice in situated social environments persist also in VR, while also showing how VR is turning into a valuable resource for studying intergroup attitudes and their change through intergroup contact. The distinguishing features of immersiveness, body ownership and embodiment provide VR with a considerable potential for stimulating perspective taking, which has been shown to be an important mediator in prejudice reduction [25], and simulating highly realistic social environments.

### 5.1 Overview and future research directions

The existing literature has used either a majority perspective to intergroup contact (i.e. embodiment in an ingroup member), or minority perspective (i.e. embodiment in an outgroup member). The latter option fully exploits the distinguishing features of VR, as it allows a highly realistic experience from the perspective of a stigmatised minority member. Existing evidence is nevertheless contrasting: while studies employing the majority perspective have shown solid potential to decrease prejudice towards stigmatised minority groups, studies using the minority outgroup perspective show that embodying an outgroup member can either lead to reduction of or increase in prejudice. Studies using both explicit and implicit measures of intergroup attitudes seem to indicate that implicitly assessed biases are more likely to change from embodying an outgroup member. This might relate to the rather visceral experience of "being an outgroup member" which might serve to associate positive self-evaluation with that outgroup.

While there has been so far little attention to mediating and moderating mechanisms of prejudice reduction in VR, preliminary evidence [82,84,86] suggests that this could depend upon the affect elicited during the embodied experience, in line with earlier evidence that negative affect during intergroup encounters can increase implicit bias [104]. Living negative affect in the body of a minority member could indeed lead to withdrawal behaviour and worsened attitudes toward said minority, underlining the importance of understanding the affective determinants of intergroup attitudes. Nevertheless, the results obtained by Kishore et al. [72] seem to contrast said pathway, since embodying a Black avatar that experiences discriminatory behaviour by a White police officer was found to increase participants' helping behaviour. Thus, one potential reason for this seemingly incompatible set of findings is the nature of the chosen outcome measures: while negative experiences associated with living a minority group perspective in VR might lead to defensive tendencies to distance oneself from the minority outgroup's reality and thus negatively influence implicit outgroup evaluations, the VR experience might positively affect behaviour through other routes of processing than implicit associations, such as through moral evaluations activating various aspects of empathy.

Intervention designers therefore face a dilemma: on the one hand, they want to give majority participants an experience that reflects that of a stigmatised minority member as accurately as possible to induce empathy and moral considerations about discrimination by providing an understanding of "what it is like to be that person"; on the other hand, if the experience elicits strong negative affect, this might lead to more negative attitudes (at least on an implicit level) which can lead to *more* discriminatory behavior in the future [105]. To what degree prolonged and/or repeated exposure to embodiment interventions could also lead to explicit attitude change remains an open question but could be hypothesised from theoretical models of attitude change [106]

The fact that an intervention works differently on explicit and implicit attitudes is not unique to VR interventions (e.g. [107,108] and relates to the general divergence of implicit and explicit attitudes and their relative contribution to behaviour, a much-debated issue in (social) psychology (see e.g. [13,98–101]). It also underlines the importance of selecting outcome measures that align with specific research questions: if, for example, the main aim of an intervention is to combat discriminatory behavior, such behavior should also be the main outcome measure. However, few of the examined studies have included actual behavior as an outcome, again reflecting a larger issue in psychological science [109].

The stigmatized targets in the included studies represented a wide variety of minority groups, such as ethnic minorities, gender and sexual minorities, obese people, neurological patients, elderly people, drug users, and more. Similarly to the previously discussed perspective, the results lead to infer that regardless of the target group, using VR to embody an outgroup member can both improve intergroup attitudes and deteriorate them. The latter effect also seems to be more prevalent in research designs using embodiment in an outgroup member. The fact that VR experiences sometimes lead to more negative attitudes towards outgroups poses a significant challenge for this research field, given that interventions should always follow a "first, do no harm" principle. Identifying specific factors that contribute to deterioration of outgroup attitudes must therefore be a major focus of future research. This is also relevant from an applied perspective: designers of VR games, for example, should be aware which game features might contribute to an increase in intergroup biases.

Those studies that explicitly examined moderating variables point to the importance of two participant-level factors that should be considered in a study design: on the one hand, participants differ on traits that make them more or less susceptible to effects of any prejudice-reduction intervention, such as empathy [76] or the importance they generally place on group memberships [78]. On the other hand, participants' involvement in and identification with the VR experience can contribute to more desirable effects [54]. These two factors might well be interconnected and these complexities should be considered in intervention designs. Relatedly, some emerging evidence suggests that the immersiveness of the experience may influence the effectiveness of interventions to reduce intergroup bias from the perspective of a stigmatised ethnic minority [102]. Taken together, at least part of the effect of immersiveness might thus be due to participants being better able to identify and get involved with the VR experience. In addition, the full experience of body ownership and identification with the embodied avatar seem to be critical, though understudied, moderators of the effect of VR contact on prejudice.

Despite the urge of gathering more insight on mechanisms specific to VR that could explain findings (e.g. immersion, body ownership, etc.), it is undeniable that this emerging method has great potential to study and reduce prejudice. Nevertheless, the evidence collected up to date calls for further investigating the role of affect in influencing changes in attitudes when embodying an outgroup member. Indeed, while empathy and perspective-taking have established roles as emotional and cognitive mediators in prejudice reduction, there are also other affective and identity-related factors that have been seen to have a powerful influence on the

contact effects on intergroup attitudes such as intergroup anxiety, threats, morality, contact motivation and others (e.g. [25]) and that need to be included also into VR contact paradigms.

## 5.2 Methodological issues and advantages

Considering the novelty of VR as a method to investigate and act on prejudice, there is still great heterogeneity and discordance on the best practices to adopt. First and foremost, the majority of the included studies (n = 34) have not controlled for the successfulness of the VR experience in terms of immersion, body ownership, or spatial presence. Given the centrality of such mechanisms in ensuring the illusion of being there [35] and perception of the virtual body as the subject's own [37], the absence of such experimental checks is a considerable limitation.

The variability of methods and lack of clarity in the operationalization of embodiment is also an emerging issue, as a significant number of studies do not provide the participants with a virtual body, but limit the VR experience to a "first person point of view", regardless of the degree of interactivity allowed in the design. Whereas this is usually the case with studies using 360˚ videos, it sometimes occurs in fully computer-designed environments as well (e.g. [81,96]). Given that not owning an avatar undermines feelings of body ownership by definition, it is a particularly important issue, especially in case of interventions based on embodying an outgroup member.

Rather strikingly, just one of the included studies [90] used technology to create VR-mediated contact between avatars steered by real members of different groups. Instead, the studies reviewed here have largely focused on interactions with computer-controlled virtual agents or avatars controlled by an experimenter. VR would seem like an ideal extension of computer-mediated or e-contact [31] that would not be limited to e.g. text-based contact but would be much closer to actual, real-world contact between majority and minority group members. It is well possible that intergroup contact in VR equalises status between groups, as seems to be the case in computer-mediated contact [30]. Few studies, however, have measured or even taken into consideration Allport's positive contact conditions [23], namely equal status, shared goals, intergroup cooperation, and support by institutions or authorities. Without said conditions, interventions aimed at reducing prejudice through positive intergroup contact in VR would fail at eventually enabling positive contact, diminishing the potential to obtain positive contact effects or even laying the foundations for negative contact effects to occur. Future studies should therefore explore VR's potential to create optimal conditions for intergroup contact to reduce prejudice [24].

Moreover, as previously observed by [43], there is a general underuse of physiological and neurological measures on the one hand, and behavioural measures on the other hand, in favour of self-reported ones. To provide a more robust test of their interventions and to overcome limitations related to social desirability of explicit measures of prejudice, many studies have indeed complemented explicit measures with the measures of implicit attitudes such as the IAT, which we discussed in the section above.

As a final remark, the amount of detail reported is generally insufficient when it comes to experimental procedures, such that it was hardly possible to evaluate the degree of bias (see Appendix 1). Further, pre-registration was rare, making an assessment of possible outcome omissions impossible. On a positive note, recent studies tended to employ more sophisticated methods than earlier examples, indicating that the field is moving from initial proof-of-concept and pilot studies to more rigorous, systematic evaluations of interventions aimed at reducing prejudice.

## Supporting information

**S1 Checklist.**
(DOC)

**S1 Appendix.**
(TIF)

**S2 Appendix.**
(PDF)

## Author Contributions

**Conceptualization:** Matilde Tassinari, Matthias Burkard Aulbach.

**Data curation:** Matilde Tassinari, Matthias Burkard Aulbach, Inga Jasinskaja-Lahti.

**Formal analysis:** Matilde Tassinari, Matthias Burkard Aulbach.

**Funding acquisition:** Inga Jasinskaja-Lahti.

**Investigation:** Matilde Tassinari.

**Methodology:** Matilde Tassinari, Matthias Burkard Aulbach.

**Project administration:** Inga Jasinskaja-Lahti.

**Supervision:** Inga Jasinskaja-Lahti.

**Writing – original draft:** Matilde Tassinari, Matthias Burkard Aulbach, Inga Jasinskaja-Lahti.

**Writing – review & editing:** Matilde Tassinari, Matthias Burkard Aulbach, Inga Jasinskaja-Lahti.

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
