## [Decision Letter · Decision Letter 0]

5 Dec 2021

PONE-D-21-31505The use of virtual reality in studying prejudice and its reduction: a systematic reviewPLOS ONE

Dear Dr. Tassinari,

Thank you for submitting your manuscript to PLOS ONE. After careful consideration, we feel that it has merit but does not fully meet PLOS ONE’s publication criteria as it currently stands. Therefore, we invite you to submit a revised version of the manuscript that addresses the points raised during the review process.

We look forward to receiving your revised manuscript.

Kind regards,

Michelangelo Vianello, Ph.D.

Academic Editor

PLOS ONE

Additional Editor Comments:

Dear Author,

Thank you for submitting your revised manuscript to PLOS ONE.

I have now received two independent reviews who are quite consistent regarding what should be fixed before publishing the manuscript. The two reviews are clear, and I agree with all comments and suggestions. I do think that you can effectively and thoroughly address all issues in a revision, so I invite you to submit a revised version of the manuscript. Please include both a clean version of the revised manuscript and a track changes version during the next submission. Also include a cover letter. If you wish to write a rebuttal for some of the reviewers’ suggestions rather than revising the manuscript, please add them in the cover letter.

After reading the reviews and the manuscript, I would like to focus the authors on three important issues that needs improvement.

Research Question.

R1 highlighted that the first part of the manuscript should be integrated with a summary of the social cognition literature on the malleability of attitudes and stereotypes. The authors will realize that doing this will also help contextualizing the role of emotion valence in the moderation of the direction of change after contact. Also, adding a good background will help identifying important questions in the literature that this systematic review might help to answer. Indeed, I would like to specifically focus the authors on this point: rather than answering a generic like “Is VR effective in changing implicit and explicit attitudes and stereotypes?”, more specific and more useful questions could be answered with this review. For instance: Under which conditions virtual contact leads to changes in implicit and explicit attitudes and stereotypes? Is the effect of real vs virtual contact the same? Do automatic attitude, prejudice or stereotype change less than their explicit counterparts? Answering specific, narrow and theoretically important questions like these is the final goal of a systematic review.

All relevant studies located?

Both I and R1 have been very surprised that the query used to locate articles did not include the word “attitude”. I think this should be fixed because neglecting all studies that investigated the effect of virtual contact on attitude rather than prejudice introduces a strong bias.

Quality of research systematically appraised.

R2 suggested that readers may not find what they look for. The core components of a systematic review as opposed to a classic -although thorough- narrative review may not be present in the current version of the manuscript. Narrative reviews are less useful for guiding policy or decision making. The style in which the results are summarized is indeed quite similar to that of narrative reviews, in which an expert opinion is discussed in terms of the existing literature. A systematic review should not only unbiasedly synthesize all studies on a specific research topic, but also critically evaluate them, and weight their informational value accordingly. For instance, the research question that leads this project is about effectiveness (does the intergroup contact created in VR reduce prejudice toward stigmatized groups?). As any other systematic review on effectiveness, randomized controlled trial are the golden rule. Other study designs should have less weight while summarizing the results. The categorization of studies according to their methodological soundness and related informational value should be crystal clear. For instance, the presence of random subject assignment, control groups, and other critical design issues such as type of contact (positive vs negative) or type of outcome (e.g., affective, cognitive, or behavioral components of attitudes) should be included in Table 1 and -above all- used by the authors to determine informational value of the studies while synthesizing the results. Hopefully, this may lead to a less ambiguous scenario than the current one in which basically every set of studies that lead to the same conclusion is contrasted with other studies that lead to different conclusions.

Minor

Plos One does not copyedit the manuscript before publication, so I do suggest getting the article revised by a native speaker. There are some typos that should be fixed (e.g. “methodical” in the abstract; “too” p. 16; “16” p. 17).

Yours sincerely,

Michelangelo Vianello

Reviewers' comments:

Reviewer's Responses to Questions

**Comments to the Author**

1. Is the manuscript technically sound, and do the data support the conclusions?

Reviewer #1: Partly

Reviewer #2: Partly

2. Has the statistical analysis been performed appropriately and rigorously? 

Reviewer #1: N/A

Reviewer #2: N/A

3. Have the authors made all data underlying the findings in their manuscript fully available?

Reviewer #1: No

Reviewer #2: Yes

4. Is the manuscript presented in an intelligible fashion and written in standard English?

Reviewer #1: Yes

Reviewer #2: Yes

5. Review Comments to the Author

Reviewer #1: Title: The use of virtual reality in studying prejudice and its reduction: a systematic review

The present review illustrates VR studies published to date on social attitudes and stereotyping with the goal of highlighting the contributions that VR techniques can offer in this field as well as the limitations of the current work.

The goal of this review is very interesting and timely since the use of VR techniques is rapidly growing in social psychology research. The VR techniques indeed represent a promising (but still under-investigated instrument) in social research concerning intergroup dynamics.

However, I have some reservations about the organization, rationale, and theoretical background of this review that prevent me from supporting its publication in the present form (see my comments/suggestions to authors).

Introduction:

Authors do not provide a background of the topic. This review is on social attitudes and stereotypes, and their malleability. I'm actually quite surprised to see that the reference list includes only a few studies from a Social Psychology or Social Cognition journal, despite the fact that this field has been studying the topic for several decades. I'm not trying to be a gatekeeper here as I recognize that many fields beyond Social Psychology/Cognition are interested in this topic. But you can't simply ignore 30+ years of scholarship. What do we know about social attitudes and stereotyping? How can we assess them? How do they operate? How can they be changed?

Similarly, I suggest introducing and describing some key concepts and terms that are essential to understand the topic (e.g., social groups, attitude, stereotype, intergroup relations, group membership, in-group and out-group members). As regards the terms included in the present version of the manuscript, I have some reservations about the use of the term “prejudice”. The term “prejudice” typically refers to negative evaluations that may be preconceived and consciously experienced. The studies reported in this review included beliefs (stereotypes) and evaluations with some degree of favor or disfavor (attitudes) about social groups assessed by using different paradigms (e.g., explicit and implicit measures). I would suggest the authors to substitute the term “prejudice” with the terms attitudes and stereotypes.

In addition, I suggest the authors to better highlight the motivation for using VR in this field. Over the past decades, several studies have attempted to develop interventions (e.g., see Dasgupta and Greenwald, 2001; Dasgupta, DeSteno, Williams, & Hunsinger, 2009; Legault, Gutsell, & Inzlicht, 2011; Mann & Kawakami, 2012; Lai et al. 2014, 2016) or used specific techniques (e.g., Marini et al. 2018) to produce changes in attitudes and stereotypes. How VR can help us to understand social interactions? Surely there is a valid motivation for doing this, right? What important gap in the social attitudes and stereotypes literature does this project fill? By providing the relevant literature on this topic, you might specify this gap to the readers in order to understand the potential and (maybe) unique contribution that VR can provide.

Results:

Authors illustrate the contribution of VR by considering the type of contact, stigma, presence of mediators, and moderators in the different studies. I think that authors should also create a section in which they compare the results found using implicit and explicit measures of attitudes/stereotypes or other instruments to assess relevant psychological constructs (e.g., empathy) in social relationships. It is important for the readers to be able here to identify the results obtained using different outcome measures. At the moment, these results are not clearly presented in the manuscript.

Conclusion and Discussion:

Please clarify also the effects of VR based on the outcome measures used in the study. What are the conclusions about implicit measures? What are the conclusions about explicit measures? What about the other paradigms used in these studies?

Minor points:

- Page numbers are missing.

- Please provide a description of the experimental paradigms mentioned in the text (e.g., shooting paradigm, Implicit Association Test).

- Please provide also in the main manuscript the main terms of research used in this review. I think that would have been appropriate include in your search also the term “attitude” and the most used instruments in this field (e.g., Implicit Association Test).

- Page 19 (?): Please consider some additional studies about the relation between implicit-explicit measures (e.g., Nosek and Smyth, 2007; Nosek et al., 2007; Greenwald and Nosek, 2008; Cunningham et al. 2004)

- Page 21(?): “Nevertheless, some contrasting evidence shows that embodying black avatars can

also lead to worsened implicit attitudes, depending on the social context in which the experience takes place (Banakou et al., 2020)”

Can you clarify this claim in the text? Which social contexts were examined in this study? Which social context worsened implicit attitudes?

Reviewer #2: Let me begin by saying that I learned something from reading this manuscript and I don’t have a lot to criticize. My summary is simple (though I’ll elaborate): I’m not certain that this is an area of investigation that is ready yet for a meta-analysis. In brief, the manuscript reads more like a very thorough Introduction than a meta-analysis. Often times, small groupings of results are pitted against one another such as in the following section (p. 19): “Similar encouraging results have been found when measuring explicit attitudes, as shown by Christofi et al. (2020) and Tong et al. (2020, 2017). However, other studies contest these findings by showing that intergroup contact experienced from the minority perspective doesn’t necessarily have any effect on intergroup attitudes (see Hasler et al., 2016; Oh et al., 2016; Starr et al., 2019).” This, to me, reads like an Introduction wherein the author(s) would include their preferred interpretation as well as a “but see” section of the citation. That said, Section 4.3 (Methodological issue and advantages) begins to outline what remains to be done. In so doing, this manuscript could serve well as a summary of the state of the field as well as an initial attempt at setting up the next couple of years of research. The quibble that the outline may be premature by those couple of years is obviously based on a somewhat arbitrary decision (i.e., exactly how many studies are needed before we are ready for a meta-analysis?). In the end, I found this manuscript more useful in terms of how it highlights what hasn’t been done. That is, of course, somewhat atypical for, as the title implies, a “systematic review”. As a reader, I might feel as though I had not gotten what I came for. I wonder if a change in title might be in order. Something like “a systematic review of the early research returns and an outline for the near future”. Something that both conveys that 1) there is not a lot of research completed yet (and so the conclusions will be relatively few and tenuous) and 2) the reader will get an organized view of what needs to happen next. To that end, I think the authors could spend a bit of time focusing current section 4.3 toward future directions and adding a bit more about what is needed. In that way, in my opinion, this would become something that anyone interested in entering into this subarea should read.

6. PLOS authors have the option to publish the peer review history of their article (what does this mean?). If published, this will include your full peer review and any attached files.

Reviewer #1: No

Reviewer #2: No

---

## [Author Response · Author response to Decision Letter 0]

18 Feb 2022

EDITOR COMMENTS

Dear Dr Vianello,

We are submitting the revised version of our previously submitted manuscript “The use of virtual reality in studying prejudice and its reduction: a systematic review”, manuscript-ID: PONE-D-21-31505. We would like to thank both reviewers and yourself for the valuable comments on the earlier version of the manuscript and we revised the text accordingly.

Specifically, the text now gives a more thorough theoretical introduction into attitudes, prejudice, and their change as traditionally studied in social psychological research and in VR settings in particular. Relatedly, we have updated and expanded our search with the terms “outgroup attitude” and “intergroup attitude” to ensure retrieving all relevant literature.

Further, the results and discussion section now follow a more systematic approach by better acknowledging the quality of the discussed research: it provides an overview of the whole field while putting a particular emphasis on the most reliable and best executed studies. Accordingly, we place stronger emphasis on what we consider to be key articles in the field, which also helps to see consensus and possible contradictions in research and thus provide clearer answers to more specific questions.

Please find our responses to the reviewers’ specific comments in the attached “response to reviewers” file.

We hope that you consider these revisions as positive and the revised manuscript eligible for publication in PLOS ONE. 

Response to reviewer comments

Reviewer #1: Title: The use of virtual reality in studying prejudice and its reduction: a systematic review

The present review illustrates VR studies published to date on social attitudes and stereotyping with the goal of highlighting the contributions that VR techniques can offer in this field as well as the limitations of the current work.

The goal of this review is very interesting and timely since the use of VR techniques is rapidly growing in social psychology research. The VR techniques indeed represent a promising (but still under-investigated instrument) in social research concerning intergroup dynamics.

However, I have some reservations about the organization, rationale, and theoretical background of this review that prevent me from supporting its publication in the present form (see my comments/suggestions to authors).

Introduction:

Authors do not provide a background of the topic. This review is on social attitudes and stereotypes, and their malleability. I'm actually quite surprised to see that the reference list includes only a few studies from a Social Psychology or Social Cognition journal, despite the fact that this field has been studying the topic for several decades. I'm not trying to be a gatekeeper here as I recognize that many fields beyond Social Psychology/Cognition are interested in this topic. But you can't simply ignore 30+ years of scholarship. What do we know about social attitudes and stereotyping? How can we assess them? How do they operate? How can they be changed?

We thank the reviewer for pointing out the overlook of the theoretical and methodological advances in a study on intergroup attitudes and prejudice in social psychology, which compromised the development of a solid foundational background of this study. However, as being social psychologists ourselves, we indeed agree that this is critical not only in terms of presenting the point of departure for the readership but also for the success of the review to describe and critically evaluate the research on intergroup contact and attitudes in VR. We have now provided a better theoretical overview of the concepts studied with more references to social psychological literature on the topic.

Similarly, I suggest introducing and describing some key concepts and terms that are essential to understand the topic (e.g., social groups, attitude, stereotype, intergroup relations, group membership, in-group and out-group members). As regards the terms included in the present version of the manuscript, I have some reservations about the use of the term “prejudice”. The term “prejudice” typically refers to negative evaluations that may be preconceived and consciously experienced. The studies reported in this review included beliefs (stereotypes) and evaluations with some degree of favor or disfavor (attitudes) about social groups assessed by using different paradigms (e.g., explicit and implicit measures). I would suggest the authors to substitute the term “prejudice” with the terms attitudes and stereotypes.

Again, we agree that some key concepts were not introduced sufficiently and have now elaborated on this. We also agree that the need for more specificity particularly relates to the concept of prejudice in relation to other concepts such as intergroup/outgroup (implicit/explicit) attitudes, ingroup/intergroup bias and intergroup/group stereotypes and emotions. Though often being studied and operationalised interchangeably, these concepts nevertheless refer to different components and forms of prejudice and they have been studied with different measures, which on their own part, also sets the limits for studying and postulating prejudice, as noted by the reviewer in the case of IAT measures used to study implicit bias. In the revised version of the manuscript, we not only put efforts in explaining the differences between the concepts and measures but also pay closer attention to the used concepts and measures in evaluating the results of the reviewed studies. 

In addition, I suggest the authors to better highlight the motivation for using VR in this field. Over the past decades, several studies have attempted to develop interventions (e.g., see Dasgupta and Greenwald, 2001; Dasgupta, DeSteno, Williams, & Hunsinger, 2009; Legault, Gutsell, & Inzlicht, 2011; Mann & Kawakami, 2012; Lai et al. 2014, 2016) or used specific techniques (e.g., Marini et al. 2018) to produce changes in attitudes and stereotypes. How VR can help us to understand social interactions? Surely there is a valid motivation for doing this, right? What important gap in the social attitudes and stereotypes literature does this project fill? By providing the relevant literature on this topic, you might specify this gap to the readers in order to understand the potential and (maybe) unique contribution that VR can provide.

We agree with the reviewer’s comment that the motivation to use VR in the field of intergroup processes is not self-evident despite the rapid development of VR technologies and the increase in intervention studies using this environment. We now have added sections to the introduction that make clear why VR has been seen as well-suited for this field of research and how it relates to and expands other lines of prejudice research.

Results:

Authors illustrate the contribution of VR by considering the type of contact, stigma, presence of mediators, and moderators in the different studies. I think that authors should also create a section in which they compare the results found using implicit and explicit measures of attitudes/stereotypes or other instruments to assess relevant psychological constructs (e.g., empathy) in social relationships. It is important for the readers to be able here to identify the results obtained using different outcome measures. At the moment, these results are not clearly presented in the manuscript.

Upon revising our manuscript, following the recommendation of Reviewer 2, we have now shifted the focus of the results section to what we consider key high quality studies in this field and report their results in more detail (as opposed to other studies that suffer from serious methodological problems). This includes an enhanced focus on effects on different kinds of outcomes, as proposed by the reviewer.

Conclusion and Discussion:

Please clarify also the effects of VR based on the outcome measures used in the study. What are the conclusions about implicit measures? What are the conclusions about explicit measures? What about the other paradigms used in these studies?

The discussion section now provides more clarification on the different studied outcomes, also distinguishing the effects on implicit from those on explicit measures, and again, with a special focus on key studies in the field. 

Minor points:

- Page numbers are missing.

Page numbers have been added to the manuscript.

- Please provide a description of the experimental paradigms mentioned in the text (e.g., shooting paradigm, Implicit Association Test).

In the original version of the manuscript, we described the IAT in paragraph 4.1. In the revised version, we have now introduced the IAT in the theoretical background section, and provide a short description of the shooter bias in result section 4.2.2.

- Please provide also in the main manuscript the main terms of research used in this review. I think that would have been appropriate include in your search also the term “attitude” and the most used instruments in this field (e.g., Implicit Association Test).

The text now includes a footnote with the search terms. We agree that the term “attitude” is important. However, including the term “attitude” would have led to an insurmountable amount of search results unrelated to the field of intergroup relations. We therefore expanded our search with the terms “outgroup attitude” and “intergroup attitude”. This brought 19 new studies in our review.

- Page 19 (?): Please consider some additional studies about the relation between implicit-explicit measures (e.g., Nosek and Smyth, 2007; Nosek et al., 2007; Greenwald and Nosek, 2008; Cunningham et al. 2004)

Obviously, the relation between implicit and explicit measures is a very debated issue, and we do agree that a more in-depth discussion of the differences between the implicit measures based on association strengths and explicit measures reflecting normative evaluative processing is in place in order to better understand the potentially different effects of VR contact in different studies and the potential on VR in studying and shaping implicit and explicit attitudes. We thank for the suggested references as they provide important arguments to this discussion (see paragraphs 1.1, 4.2.1, and 5.1).

- Page 21(?): “Nevertheless, some contrasting evidence shows that embodying black avatars can

also lead to worsened implicit attitudes, depending on the social context in which the experience takes place (Banakou et al., 2020)”

Can you clarify this claim in the text? Which social contexts were examined in this study? Which social context worsened implicit attitudes?

This study is now discussed in more detail in paragraphs 4.2.1 and 4.2.2, as we regard it to be one of the key studies in this field.

REVIEWER #2: 

Let me begin by saying that I learned something from reading this manuscript and I don’t have a lot to criticize. My summary is simple (though I’ll elaborate): I’m not certain that this is an area of investigation that is ready yet for a meta-analysis. In brief, the manuscript reads more like a very thorough Introduction than a meta-analysis. Often times, small groupings of results are pitted against one another such as in the following section (p. 19): “Similar encouraging results have been found when measuring explicit attitudes, as shown by Christofi et al. (2020) and Tong et al. (2020, 2017). However, other studies contest these findings by showing that intergroup contact experienced from the minority perspective doesn’t necessarily have any effect on intergroup attitudes (see Hasler et al., 2016; Oh et al., 2016; Starr et al., 2019).” This, to me, reads like an Introduction wherein the author(s) would include their preferred interpretation as well as a “but see” section of the citation. That said, Section 4.3 (Methodological issue and advantages) begins to outline what remains to be done. In so doing, this manuscript could serve well as a summary of the state of the field as well as an initial attempt at setting up the next couple of years of research. The quibble that the outline may be premature by those couple of years is obviously based on a somewhat arbitrary decision (i.e., exactly how many studies are needed before we are ready for a meta-analysis?). In the end, I found this manuscript more useful in terms of how it highlights what hasn’t been done. That is, of course, somewhat atypical for, as the title implies, a “systematic review”. As a reader, I might feel as though I had not gotten what I came for. I wonder if a change in title might be in order. Something like “a systematic review of the early research returns and an outline for the near future”. Something that both conveys that 1) there is not a lot of research completed yet (and so the conclusions will be relatively few and tenuous) and 2) the reader will get an organized view of what needs to happen next. To that end, I think the authors could spend a bit of time focusing current section 4.3 toward future directions and adding a bit more about what is needed. In that way, in my opinion, this would become something that anyone interested in entering into this subarea should read.

We thank the reviewer for their thoughtful comments on the manuscript and would first like to clarify that this article does not attempt to provide a meta-analysis in the sense of statistically weighing the evidence - we agree that research in this field is too under-developed and unsystematic for this to be done in a meaningful way. Instead, we aimed at providing a systematic review of the research on VR contact and intergroup attitudes conducted so far in order to describe the different lines of research, the designs and measures used, and the effects obtained. We also agree that the way we described our results in the original version of the manuscript was somewhat dissatisfying as they did not draw a clear picture on whether VR interventions are effective or not. We agree with the reviewer that this was due to the equal weight given to the reviewed studies neglecting the clear differences in their quality. In our revision, we now attempt to alleviate this by giving more emphasis to studies that we consider to be of high methodological quality. This way, we, to some degree, avoid the ambiguity created by seemingly assigning equal weight to methodologically heterogeneous studies and manage to come to somewhat clearer conclusions.

The reviewer writes that “this manuscript could serve well as a summary of the state of the field as well as an initial attempt at setting up the next couple of years of research” - this is exactly what we are trying to do and we hope that our revisions lay this out more clearly.

Finally, what is clear is that the field of research undergoes very rapid development, which is evident in the revised manuscript. Namely, the update of the search conducted for the original version of the manuscript has brought 19 new studies in our review published after our initial submission in 2021. The increased data-base also allowed us to draw better overview of and conclusions from the results obtained in the field.

---

## [Decision Letter · Decision Letter 1]

8 Apr 2022

PONE-D-21-31505R1The use of virtual reality in studying prejudice and its reduction: a systematic reviewPLOS ONE

Dear Dr. Tassinari,

I have received two reviews of your revised manuscript that are quite contrasting. Reviewer 2 is satisfied by your revisions and suggests publication. Reviewer 1 agrees that your revisions improved the manuscript but believes that they are not sufficient to support publication and suggests rejection.

I realize that a lot of work has gone into this manuscript during the review process. Hence, I offer you the chance of providing a second revision. I carefully read the revised manuscript, and I think that suggestions made by reviewer 1 were quite easy to address. You did not write a rebuttal to R1 suggestions made in the previous round of revisions. Instead, you explicitly agreed with them. So I guess that they are not included in the revisions due to a misunderstanding.

To clarify: R1 and/or I suggested to:

Add one or two paragraphs summarizing the 30+ years of research in social psychology on the malleability of attitudes and stereotypes. The summary should not be exhaustive. Yet, it should give readers a view of the great amount of research that has been conducted, framing intergroup contact into a larger set of interventions that have been extensively studied. Intergroup contact is should be presented as one of these interventions, not even the most effective (Lai et al. 2014; Lai et al. 2016).Disaggregate both the literature review in (see previous point) and your results by outcome measure: implicit and explicit attitudes were shown to be differently malleable. Also discuss later in the manuscript the implicit/explicit distinction as a moderator.

Minor

Some typos still need to be fixed (e.g. “methodical” in the abstract). This issue was found in the previous version as well.p. 5: it may be that the reader would understand that neuroimaging is subtly proposed as a more reliable substitute of indirect measures. This approach is even more evident in the discussion (p. 32). I do think that this should be avoided because most if not all experts in the field would not agree with that. These measures avoid self-reports, but this is probably everything they have in common. Also, there is no evidence at all that one is more reliable than the other. There is a famous paper by Bennett et al. (2009) in which the authors observed cerebral activity in a dead salmon…   fix the citation for the IAT.I typically refrain from this kind of suggestions, but I do think that omitting Bar-Anan & Vianello (2018) when citing Schimmack (2019) and the debate on the distinction between implicit and explicit constructs would seriously misrepresent the available empirical evidence on this topic. Please submit your revised manuscript by May 23 2022 11:59PM. If you will need more time than this to complete your revisions, please reply to this message or contact the journal office at plosone@plos.org. Please include the following items when submitting your revised manuscript:A rebuttal letter that responds to each point raised by the academic editor and reviewer(s). You should upload this letter as a separate file labeled 'Response to Reviewers'.A marked-up copy of your manuscript that highlights changes made to the original version. You should upload this as a separate file labeled 'Revised Manuscript with Track Changes'.An unmarked version of your revised paper without tracked changes. You should upload this as a separate file labeled 'Manuscript'.If applicable, we recommend that you deposit your laboratory protocols in protocols.io to enhance the reproducibility of your results. Protocols.io assigns your protocol its own identifier (DOI) so that it can be cited independently in the future. For instructions see: https://journals.plos.org/plosone/s/submission-guidelines#loc-laboratory-protocols. Additionally, PLOS ONE offers an option for publishing peer-reviewed Lab Protocol articles, which describe protocols hosted on protocols.io. Read more information on sharing protocols at https://plos.org/protocols?utm_medium=editorial-email&utm_source=authorletters&utm_campaign=protocols.

We look forward to receiving your revised manuscript.

Kind regards,

Michelangelo Vianello, Ph.D.

Academic Editor

PLOS ONE

Journal Requirements:

Reviewers' comments:

Reviewer's Responses to Questions

**Comments to the Author**

1. If the authors have adequately addressed your comments raised in a previous round of review and you feel that this manuscript is now acceptable for publication, you may indicate that here to bypass the “Comments to the Author” section, enter your conflict of interest statement in the “Confidential to Editor” section, and submit your "Accept" recommendation.

Reviewer #1: (No Response)

Reviewer #2: (No Response)

2. Is the manuscript technically sound, and do the data support the conclusions?

Reviewer #1: No

Reviewer #2: Yes

3. Has the statistical analysis been performed appropriately and rigorously? 

Reviewer #1: N/A

Reviewer #2: Yes

4. Have the authors made all data underlying the findings in their manuscript fully available?

Reviewer #1: (No Response)

Reviewer #2: Yes

5. Is the manuscript presented in an intelligible fashion and written in standard English?

Reviewer #1: Yes

Reviewer #2: Yes

6. Review Comments to the Author

Reviewer #1: I think the present version of the manuscript has improved compared to the previous one. However, I still have some concerns that prevent me from supporting its publication. Unfortunately, the authors did not fully address some crucial points raised in my previous review.

Here are some examples.

I suggested the authors to provide a background of the topic that can summarize what we know about the malleability of attitudes and stereotypes. This piece of information is still lacking in the text. In the revised manuscript, the authors only describe in more detail one kind of intervention developed to change social attitudes and stereotypes (i.e., intergroup contact). Again, you cannot ignore 30+ years of scholarship on this topic. Over the past decades, several studies have developed interventions to change attitudes and stereotypes (e.g., see Dasgupta and Greenwald, 2001; Dasgupta, DeSteno, Williams, & Hunsinger, 2009; Legault, Gutsell, & Inzlicht, 2011; Mann & Kawakami, 2012; Lai et al. 2014, 2016; Marini et al. 2018). One or two paragraphs on this topic are necessary for the readers to understand what we know in this field and how VR may be useful. By reading the introduction of this manuscript, it seems social psychologists used only interventions based on intergroup contact to modulate social attitudes and stereotypes. I know that intergroup contact was the most used intervention in the VR studies, but this does not imply that this intervention is the only one that can be implemented using VR or the only one that deserves to be described. On this latter point, please also consider some recent studies on implicit bias showing that intergroup contact and perspective-taking interventions are not the most effective interventions to modulate social attitudes and stereotypes (see, for example, Lai et al. 2014; Lai et al. 2016).

The readers deserve to know that, over the years, many experimental interventions have been developed in this research field. The background provided by the authors is too limited.

In addition, a crucial aspect that was not addressed is the effectiveness of VR on implicit and explicit measures. What do we know about that? Is VR more effective on the implicit or explicit bias? Are the results of the reviewed studies unclear on this topic? This issue is something that the authors need to raise in their review, considering that attitudes and stereotypes assessed by implicit measures have shown to predict behavior more accurately than explicit measures in some specific domains. This aspect is relevant because the final goal of reducing attitudes and stereotypes is to reduce their effect on behavior.

Again, I still have some concerns about using the term “prejudice” in this manuscript as it includes studies in which they used implicit measures as the IAT. I believe this term is not appropriate given the long debate in the literature about what implicit measures assess.

Finally, please report the correct references when you cite an instrument. The appropriate citation for the IAT is not Greenwald & Banaji (1995) but Greenwald, McGhee, and Schwartz (1998).

Reviewer #2: I read the revised manuscript with interest and pleasure. I feel as though the review process made this paper quite a bit stronger. Particularly as a result of the authors’ openness to doing a lot of additional work in response to reviewer requests. And, as the authors note in their response to the reviews, the current version is also strengthened by the number of additional papers that could now be included – the number of which serves to indicate how important the topic is. This is a paper that I will send to my students as a way to understand the state of a topic. That’s a rather self-centered compliment, but it’s the best way I know to indicate a paper’s potential utility.

I note that it will be worthwhile to do one more very slow read through for typos and things related to some annoying idiosyncrasies of the English language, particularly related to verb tenses and singular vs. plural formations.

7. PLOS authors have the option to publish the peer review history of their article (what does this mean?). If published, this will include your full peer review and any attached files.

Reviewer #1: No

Reviewer #2: No

---

## [Author Response · Author response to Decision Letter 1]

20 May 2022

Response to reviewer comments

Reviewer #1: Title: The use of virtual reality in studying prejudice and its reduction: a systematic review

I think the present version of the manuscript has improved compared to the previous one. However, I still have some concerns that prevent me from supporting its publication. Unfortunately, the authors did not fully address some crucial points raised in my previous review.

We are sorry to read that reviewer 1 did not find the previous version of the manuscript as satisfying, but it is good to know that they appreciated the volume and direction of changes made. In the current revision, we did our best to respond and correct every remaining detail of their critical comments, which, in our view, again clearly improved the potential contribution of the study.

Here are some examples.

I suggested the authors to provide a background of the topic that can summarize what we know about the malleability of attitudes and stereotypes. This piece of information is still lacking in the text. In the revised manuscript, the authors only describe in more detail one kind of intervention developed to change social attitudes and stereotypes (i.e., intergroup contact). Again, you cannot ignore 30+ years of scholarship on this topic. Over the past decades, several studies have developed interventions to change attitudes and stereotypes (e.g., see Dasgupta and Greenwald, 2001; Dasgupta, DeSteno, Williams, & Hunsinger, 2009; Legault, Gutsell, & Inzlicht, 2011; Mann & Kawakami, 2012; Lai et al. 2014, 2016; Marini et al. 2018). One or two paragraphs on this topic are necessary for the readers to understand what we know in this field and how VR may be useful. By reading the introduction of this manuscript, it seems social psychologists used only interventions based on intergroup contact to modulate social attitudes and stereotypes. I know that intergroup contact was the most used intervention in the VR studies, but this does not imply that this intervention is the only one that can be implemented using VR or the only one that deserves to be described. On this latter point, please also consider some recent studies on implicit bias showing that intergroup contact and perspective-taking interventions are not the most effective interventions to modulate social attitudes and stereotypes (see, for example, Lai et al. 2014; Lai et al. 2016).

The readers deserve to know that, over the years, many experimental interventions have been developed in this research field. The background provided by the authors is too limited.

RE: We thank the reviewer for spelling out this comment and fully agree that there was space and function for the suggested addition. In light of the reviewer’s comment, we added three paragraphs on the malleability of intergroup attitudes to provide an accurate overview about the development of intervention research in the area (see section 1.2. Malleability of intergroup attitudes). To be able to cover the volume of intervention research, we decided to focus on meta-analytic studies. Thus, in the first paragraph, we describe Paluck and Green’s (2009) classification of social scientific and psychological interventions designed to influence intergroup bias. In paragraphs 2 and 3, we discuss the malleability of implicit racial prejudice based on the results of the meta-analyses of Lai et al. (2014, 2016), followed by the results of Beelmann and Heinemann’s (2014) meta analysis of different structured programs to reduce explicit prejudice and negative intergroup attitudes. Only then do we move on to discussing the interventions based on contact theory. 

In addition, a crucial aspect that was not addressed is the effectiveness of VR on implicit and explicit measures. What do we know about that? Is VR more effective on the implicit or explicit bias? Are the results of the reviewed studies unclear on this topic? This issue is something that the authors need to raise in their review, considering that attitudes and stereotypes assessed by implicit measures have shown to predict behavior more accurately than explicit measures in some specific domains. This aspect is relevant because the final goal of reducing attitudes and stereotypes is to reduce their effect on behavior.

RE:Upon revising our manuscript, we have now added a paragraph in the “Results” section to outline differences in findings regarding the effect of VR contact on intergroup bias assessed using implicit and explicit measures. Based on our analysis, we now conclude that although results are somewhat inconsistent for both measures and depend on the type of intergroup contact studied (majority vs minority perspective), “It seems like implicit measures unveil potential effects of VR-based interventions that might not appear in explicit measures of intergroup attitudes” (p. 27). By making this conclusion, we agree with the reviewer that this is an important conclusion to be made which does not only highlight the potential of VR contact to improve (implicit) attitudes but also outlines its potential to change intergroup behaviors. Furthermore, we have expanded those findings further in the discussion. 

Again, I still have some concerns about using the term “prejudice” in this manuscript as it includes studies in which they used implicit measures as the IAT. I believe this term is not appropriate given the long debate in the literature about what implicit measures assess.

RE:We understand this concern. We have taken care of this point by describing the way the concepts of prejudice, intergroup attitudes and intergroup bias are separate but interrelated and how they have been used in different literatures to pinpoint the specific nature of the phenomena studied. We also carefully went through the text and checked that we refer to prejudice only when we talk about the general area of prejudice reduction intervention research. In contrast, we replaced “prejudice” with “intergroup bias” each time we discuss implicit measures. We hope that this way we can do both: speak to the readership about the importance of prejudice reduction and to acknowledge different theoretical and methodological approaches to study intergroup phenomena.

Finally, please report the correct references when you cite an instrument. The appropriate citation for the IAT is not Greenwald & Banaji (1995) but Greenwald, McGhee, and Schwartz (1998).

RE:We corrected this unintended error in referencing. 

Reviewer #2: Title: The use of virtual reality in studying prejudice and its reduction: a systematic review

I read the revised manuscript with interest and pleasure. I feel as though the review process made this paper quite a bit stronger. Particularly as a result of the authors’ openness to doing a lot of additional work in response to reviewer requests. And, as the authors note in their response to the reviews, the current version is also strengthened by the number of additional papers that could now be included – the number of which serves to indicate how important the topic is. This is a paper that I will send to my students as a way to understand the state of a topic. That’s a rather self-centered compliment, but it’s the best way I know to indicate a paper’s potential utility.

I note that it will be worthwhile to do one more very slow read through for typos and things related to some annoying idiosyncrasies of the English language, particularly related to verb tenses and singular vs. plural formations.

RE:We are pleased to read that reviewer #2 considers our work ready for publication, and we wish to thank for the good words on the meaningfulness of the manuscript. We have now proofread the manuscript to check for typos and enhance linguistic consistency. We hope that this and the additional revisions have further strengthened the manuscript.

---

## [Editor Report · Decision Letter 2]

17 Jun 2022

The use of virtual reality in studying prejudice and its reduction: a systematic review

PONE-D-21-31505R2

Dear Dr. Tassinari,

We’re pleased to inform you that your manuscript has been judged scientifically suitable for publication and will be formally accepted for publication once it meets all outstanding technical requirements.

Kind regards,

Michelangelo Vianello, Ph.D.

Academic Editor

PLOS ONE

---

## [Editor Report · Acceptance letter]

22 Jun 2022

PONE-D-21-31505R2 

The use of virtual reality in studying prejudice and its reduction: a systematic review 

Dear Dr. Tassinari:

I'm pleased to inform you that your manuscript has been deemed suitable for publication in PLOS ONE. Congratulations! Your manuscript is now with our production department. 

Kind regards, 

on behalf of

Dr. Michelangelo Vianello 

Academic Editor

PLOS ONE